# Scube2 primes Dispatched and ADAM10-mediated Shh release by recruiting HDL acceptors to the plasma membrane

J. Puschmann[1], G. Steffes[2], J. Froese[1], D. Manikowski[1], K. Ehring[1], J. Wittke[1], C. Garbers [3], S. V. Wegner [1] & K. Grobe [1] ✉

Sonic hedgehog (Shh) morphogens are lipidated proteins that firmly attach to the outer plasma membrane (PM) of the cells that produce them. The process by which Shh is solubilized requires the transmembrane protein Dispatched1 (Disp), the soluble glycoprotein Scube2, the proteolytic removal of lipidated peptide termini, and the use of soluble lipoproteins (LPPs) as Shh transporters. However, their molecular interplay remains controversial. Here, we demonstrate that A Disintegrin and Metalloproteinase 10, Scube2, and Disp act synergistically to remove Shh from the PM and transfer it to LPP acceptors. We also demonstrate physical Scube2 interactions with LPPs and that these interactions increase Shh release. Finally, we demonstrate that Scube2 strongly binds to heparan sulfate (HS) on cell surfaces. These findings reveal Scube2's previously unknown role in binding low-abundance, soluble LPP carriers for Shh and recruiting these carriers to HS-rich Shh release sites at the PM to enhance morphogen release.

A distinctive characteristic of all Hedgehog (Hh) family members is that, during their biosynthesis, they undergo autocatalytic covalent attachment of a cholesteryl moiety to the C-terminus of the 19 kDa signaling domain[1]. Additionally, Hh acyltransferase (Hhat) adds a palmitate to the free N-amino group of cholesteroylated Hhs[2], resulting in secreted dual-lipidated morphogens that remain tightly bound to the outer plasma membrane (PM) leaflet of the producing cells. Next, the dual-lipidated proteins bind to extracellular heparan sulfate (HS) proteoglycans (HSPGs)[3]. HSPGs consist of highly abundant PM-associated core proteins that are linked to linear, negatively charged HS sugar polymers. These polymers interact with many extracellular proteins, including the Hhs, and influence Hh bioactivity[4–8] by assembling lipid-modified Hh into HS-bound punctate platforms on the cell surface[3,9]. How Hh is released from these platforms is the subject of intense investigation.

One soluble protein that enhances Sonic hedgehog (Shh) release to varying degrees[10–13] is Scube2 (signal sequence, cubulin (CUB) domain, epidermal growth factor (EGF)-like protein 2)[14–17]. The Scube2 signal sequence is followed by nine EGF domains, a spacer region, a cysteine-rich domain (CRD) and a C-terminal CUB domain. In general, CUB domains and EGF domains regulate biological processes by supporting protein-protein interactions[18–20]. Of note, the spacer region separating the EGF domains from the CRD and CUB domains contains an HS binding site[21].

Another Hh release factor is the PM protein Dispatched1 (Disp)[22–25]. Disp is composed of 12 transmembrane (TM) helices and two extracellular domains, making it a member of the resistance-nodulation-division family of TM efflux pumps. Disp also contains a sterol sensing domain (SSD), which, in other SSD-containing proteins, regulates cholesterol levels in cells or their transport[26]. The SSD motif in Disp therefore suggests that it extracts the C-terminal Hh sterol to transfer it to soluble acceptors. One proposed acceptor is Scube2 in vertebrates, which is thought to chaperone dual-lipidated Shh away from the producing cells[14,15]. Another proposed acceptor for lipidated Hhs are high-density lipoproteins (HDL)[27–29] that are also soluble acceptors for peripheral cholesterol[30]. Of note, their small size of 5-10 nm makes HDL abundant not only in the circulation but also in interstitial fluids that fill the spaces between the cells in the body (the interstitium), both in the adult and during development[28]. Finally, Shh can be released by cell surface-associated proteases, called sheddases, in a terminally truncated and delipidated form[31–33]. However, this mechanism is challenged by a cryo-EM structure that revealed a hydrophobic tunnel for cholesterol export in the Hh receptor, Patched (Ptch)[34]. In the presence of dual-lipidated Shh, which was chemically extracted from the PM of transfected cells, the palmitoylated Shh N-terminus blocks this tunnel to potentially initiate signaling. This interaction is interpreted as strong evidence against Hh shedding, because removing the Hh N-terminal palmitate

[1]Institute of Physiological Chemistry and Pathobiochemistry, University of Münster, Münster, Germany. [2]Institute of Neuro- and Behavioral Biology, University of Münster, Münster, Germany. [3]Institute of Clinical Biochemistry, Hannover Medical School, Hannover, Germany. ✉e-mail: kgrobe@uni-muenster.de

during release would severely impair or eliminate signaling to Ptch. However, palmitate-dependent signaling in receiving cells requires three specific predictions to be met. First, solubilization via Disp and Scube2 leaves the Hh ligand dual-lipidated. Second, dual-lipidated soluble Shh, but not similar amounts of Shh proteins lacking their terminal peptides, will elicit robust signaling in Hh reporter cells that express Ptch. Third, the absence of cell-surface sheddases will not suppress Hh solubilization in vitro and in vivo.

In this study, we tested these specific predictions in HEK293 (human embryonic kidney) cells, and in cells that have had either Disp function or the function of the cell surface sheddases A disintegrin and metalloproteinase 10 (ADAM10, or A10) and A17 knocked out. These experiments showed that Disp and Scube2 release proteolytically processed Shh but not full-length, dual-lipidated Shh. They also revealed A10 is the primary Shh-processing sheddase in HEK293 cells, and that the *Drosophila* ortholog of A10, Kuzbanian (Kuz), contributes to Hh-regulated fly development. We also immunoprecipitated Scube2 and analyzed interacting extracellular proteins by mass spectrometry. This revealed that Scube2 binds several proteases, protease inhibitors, and ApoA1, the immobile apoprotein typical of HDL. Functional assays confirmed that Scube2 interacts with HDL. They also confirmed that Scube2 interacts with HS at the cell surface. Together, these results revealed a previously unsuspected role for Scube2 in binding low abundant soluble HDL and in recruiting the soluble carrier to the HS-decorated PM to enhance Disp- and A10-mediated Shh transfer to it. Importantly, despite lacking N-palmitate, HDL-associated Shh is highly bioactive.

## Results

### A10 sheds dual-lipidated Shh from the PM

To compensate for potential drawbacks of protein overexpression, such as missing or insufficient post-translational protein modification, we routinely co-express Shh and Hhat from the same bicistronic mRNA[16] and monitor complete Hh lipidation of the PM-associated protein by reverse phase high pressure liquid chromatography (RP-HPLC) (Fig. 1A). 36 h post-transfection, media containing 10% serum were replaced with serum-free DMEM and Shh was solubilized for 6 h. Notably, HEK293 cells were not washed between media changes, leaving residual traces of serum in the assay (referred to as "serum-depleted" in this paper). We then used SDS-PAGE and Western blotting to directly compare solubilized Shh from TCA-precipitated media (m) with the cellular Shh precursor (c) and also to compare Shh solubilization from cells expressing Disp (HEK) or not (Disp$^{-/-}$)[22] (Fig. 1A). We observed that Shh release from Disp$^{-/-}$ cells was strongly decreased when compared to HEK cells expressing Disp, and maximal Shh release always required the presence of Scube2 (Fig. 1B, Supplementary Fig. 1A shows loading and specificity controls on the same stripped blot, Fig. 1B', B" shows quantification of Disp- and Scube2-dependent Shh release). This result was consistent with the known synergistic activities of Disp and Scube2 in Shh release. However, we also observed an electrophoretic mobility shift when soluble Shh (Fig. 1B, arrowhead) was compared with the corresponding membrane-bound precursor (asterisk), suggesting that the dual-lipidated Shh is not "extracted" from the PM, as this would not have changed its electrophoretic mobility. Instead, we have previously determined that the observed mobility shift resulted from the proteolytic removal of the lipidated terminal peptide anchors during Shh release[16,27,35,36]. This prompted us to investigate which sheddase releases Shh from the cell surface.

To address this question, we analyzed Shh release from Disp-expressing and A10-expressing cells and from Disp-expressing cells rendered A10 deficient by CRISPR/Cas9 (a kind gift of C. Garbers, hereafter referred to as A10$^{-/-}$ cells). We observed strongly reduced Shh solubilization from A10$^{-/-}$ cells in serum-depleted media, even in the presence of Scube2

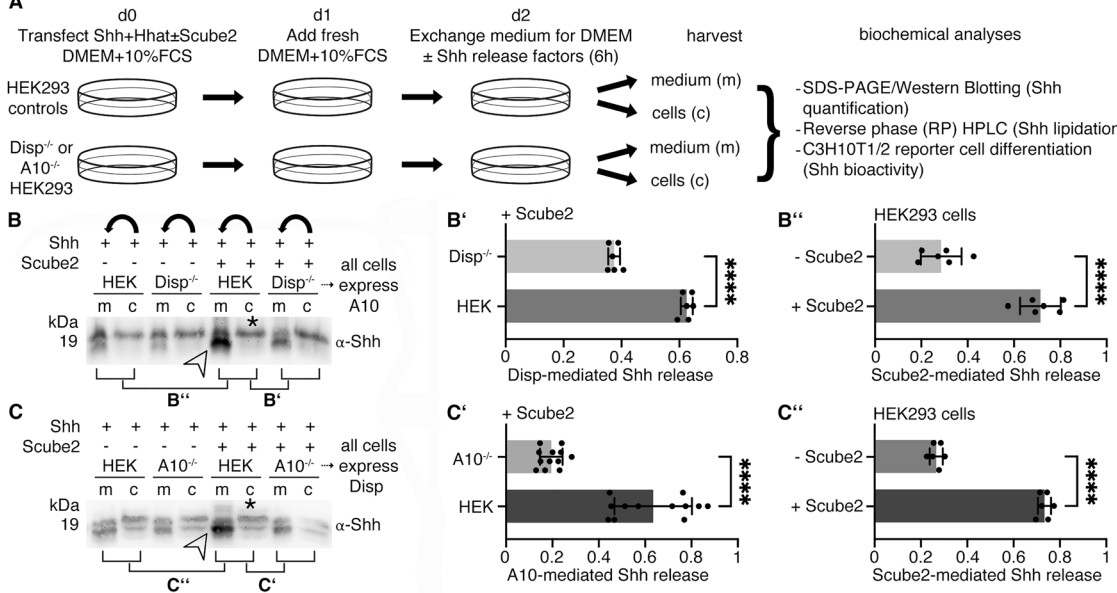

**Fig. 1 | Scube2-mediated Shh solubilization requires Disp and A10.**
**A** Experimental setup. HEK293 cells with or without Disp and A10 function were co-transfected with Shh and Hhat in the presence or absence of Scube2. Proteins secreted into the serum-depleted media were TCA precipitated and the corresponding cells were lysed for subsequent analysis by SDS-PAGE/immunoblotting or RP-HPLC. **B** Shh solubilization from Disp-expressing HEK cells (arrowhead) requires Scube2, and Shh release from Disp$^{-/-}$ cells is severely impaired. m: media, c: Shh from the corresponding cell lysate (asterisk). Bent arrows indicate Shh solubilization from cells into corresponding media in this and in all subsequent experiments. See Supplementary Fig. 1A for specificity and loading controls. **B'**) Quantification of relative Shh release from HEK and Disp$^{-/-}$ cells in the presence of Scube2 (Compared lanes are indicated in **B**). Only solubilized Shh with increased electrophoretic mobility (lower bands) was quantified. Unpaired *t* test, two-tailed. ****: *p* < 0.0001, *n* = 6. **B"**) Quantification of Shh release from Disp-expressing cells in the presence or absence of Scube2. Unpaired *t* test, two-tailed. ****: *p* < 0.0001, *n* = 6. **C** A10 and Scube2 also enhance the conversion of dually lipidated cellular Shh (asterisk) into a truncated soluble form (arrowhead). **C'**) Quantification of Shh release from A10-expressing cells and from A10$^{-/-}$ cells. Unpaired *t* test, two-tailed. ****: *p* < 0.0001, *n* = 12. **C"**) Quantification of Shh release from HEK cells expressing A10 in the presence or absence of Scube2. Unpaired *t* test, two-tailed. ****: *p* < 0.0001, *n* = 6. Error bars always represent the standard deviations of the means. See Supplementary Table 1 for detailed statistical information.

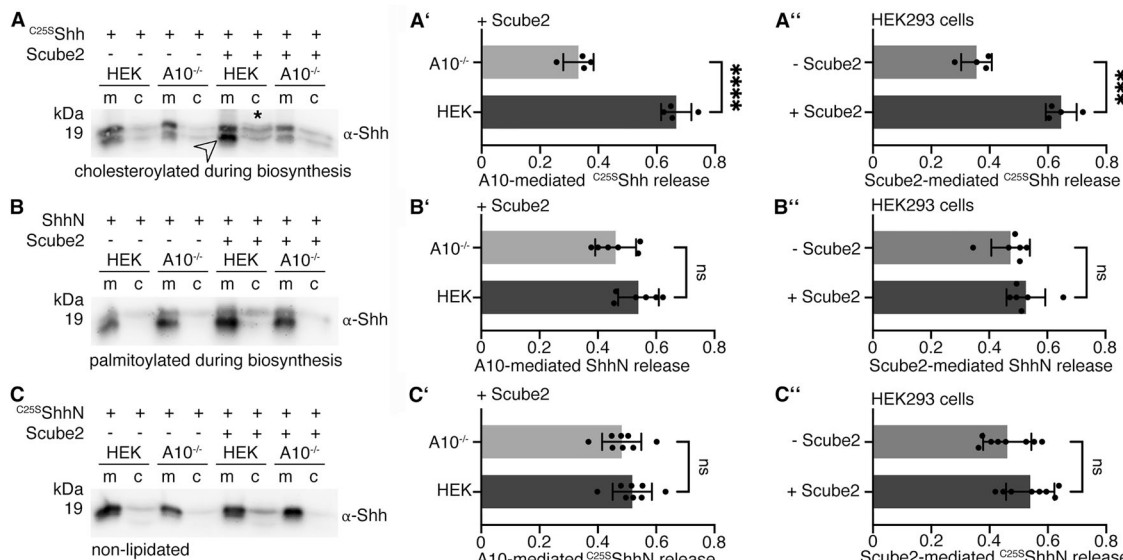

**Fig. 2 | A10 and Scube2 differentially control the release of artificial non- and monolipidated Shh variants. A–C** Solubilization of non-palmitoylated C25SShh, non-cholesteroylated ShhN, and non-lipidated C25SShhN in serum-depleted media. **A'-C')** Protein quantification from HEK cells or A10$^{-/-}$ cells in the presence of Scube2. Unpaired *t* test, two-tailed. ns: $p > 0.05$, ****: $p < 0.0001$, $n = 4$ (**A'**), $n = 6$ (**B'**), n = 8 (**C'**). **A''-C''**). Quantification of protein solubilization from HEK cells in the presence or absence of Scube2. Unpaired *t* test, two-tailed. ns: $p > 0.05$, ***: $p < 0.0002$, $n = 4$ (**A''**), $n = 6$ (**B''**), n = 8 (**C''**). Error bars always represent the standard deviations of the means. See Supplementary Table 1 for detailed statistical information.

(Fig. 1C, Supplementary Fig. 1B). This result demonstrated that Disp and Scube2 are not sufficient to release Shh from HEK293 cells, as A10 clearly contributes to this process (Fig. 1C', C''). Moreover, like Disp, A10 required Scube2 for maximal Shh release (Fig. 1C, arrowhead). As described previously[22,27], we observed an electrophoretic mobility shift of A10-solubilized Shh (Fig. 1C, arrowhead) compared to the corresponding cellular forms (asterisk), which can now be attributed to A10 activity. RP-HPLC of the solubilized material shown in Fig. 1C (indicated by the arrowhead) confirmed that A10 removed the lipidated Shh peptide termini during solubilization (Supplementary Fig. 1C). In contrast to A10, the closely related protease A17 (also called TACE) did not contribute to Shh solubilization under the same experimental conditions (Supplementary Fig. 1D).

We next examined A10 release of an artificially monolipidated C25SShh variant lacking the N-terminal palmitate and a ShhN variant lacking the C-terminal cholesterol (Fig. 2A, B, Supplementary Fig. 1E, F). We found that C25SShh release was enhanced by A10 and Scube2 (Fig. 2A, A', A''). The release of ShhN was not controlled by either A10 or Scube2 (Fig. 2B' B''), suggesting that other sheddases can cleave the ShhN N-terminus under non-specific conditions (i.e. Shh lacking dual lipidation). Indeed, the level of ShhN solubilization was similar to that of C25SShhN, an artificial non-lipidated control protein (Fig. 2C-C'', Supplementary Fig. 1G). Taken together, these results demonstrate that A10 contributes to Disp- and Scube2-mediated Shh release in vitro. They also demonstrate that the control of Shh release by A10 and Scube2 strictly depends on the presence of both Shh lipids, as previously shown for Disp and Scube2[27]. This provides an alternative explanation for the necessity of dual lipidation and its complete conservation among all Hh family members, which differs from the previously proposed roles of Hh lipidation in signaling at the Ptch level.

### Tissue-specific depletion of Hh, Disp, and the A10 ortholog Kuz results in similar phenotypes in the *Drosophila* eye

Next, we investigated the A10 ortholog Kuz and its possible contribution to Hh-regulated *Drosophila* eye development. We chose the *Drosophila* eye as a model because it consists of ~700 photoreceptors (ommatidia) that develop in a wave of Hh-regulated differentiation that moves from the posterior (p) to the anterior (a) of the eye disc, called the morphogenetic furrow (MF, Fig. 3A). Cells anterior to the MF respond to Hh released by

cells posterior to the MF by producing and releasing the same protein. This creates a cyclic, short-range Hh signaling mode that drives the furrow across the disc[27,37] and determines the number of ommatidia in the adult eye. Impaired Hh function during development therefore results in fewer ommatidia that can be easily quantified. Another advantage of studying eye development is that it is not essential for fly survival, which allowed us to avoid the deleterious pleiotropic effects of Kuz knockdown in other tissues (Supplementary Fig. 1H). We used the established eye disc-specific glass multimer reporter (GMR)-Gal4 driver[38] to express dominant-negative Kuz$^{DN}$, which suppresses endogenous Kuz in the developing eye[39], and hypothesized that this impairs the release and biofunction of endogenous Hh. While we found that GMR-controlled Kuz$^{DN}$ expression at 25 °C often resulted in pupal lethality, probably due to leaky Kuz$^{DN}$ expression outside the eye disc, we also found that surviving flies had significantly reduced numbers of ommatidia ($n = 235 \pm 22$ ommatidia/eye, GFP-expressing control discs formed eyes consisting of $n = 657 \pm 38$ ommatidia, $n = 10$ eyes were analyzed for each line, $p < 0.0001$, Fig. 3B,B'). Another established specificity control for the small eye phenotype are homozygous $hh^{bar3}$ eye discs that lack sufficient Hh expression to drive MF progression[40] ($271 \pm 28$ ommatidia, $n = 10$, $p < 0.0001$ when compared to GMR>gfp eyes, Fig. 3B,B'). This shows that impairing the activity of endogenous Kuz affects Hh-driven *Drosophila* eye development to a similar extent as the Hh$^{bar3}$ mutation[41] in a process that is unrelated to apoptosis (Supplementary Fig. 2A). Eyes made homozygous for the hypomorphic allele disp$^{S037707}$ (disp$^{LacW}$)[23] using the *Minute* technique under control of eyFLP3.5 also impaired eye development ($304 \pm 58$ ommatidia, n = 10, p < 0.0001 when compared to GMR>gfp, Fig. 3B,B'), mirroring our in vitro finding that both A10 knockout and Disp knockout impair Shh release to a similar extent (Fig. 1B, C). Finally, expression of $^{HA}$Hh (the inserted HA tag makes the N-terminal peptide protease-resistant[35,42]) under the same GMR-Gal4 control in eye discs lacking most endogenous Hh (hh$^{bar3}$/hh$^{AC}$) also resulted in a small eye phenotype ($n = 244 \pm 22$ ommatidia, $n = 10$, $p < 0.0001$ when compared to GMR>gfp, Fig. 3B,B')[27,42]. Since this very similar phenotype is caused by the inserted protease-resistant HA tag, as demonstrated by control Hh expression in the same genetic background ($n = 649 \pm 21$ ommatidia, $n = 10$, $p = 0.99$ when compared to GMR>gfp, Fig. 3B,B'), we conclude that the small eye phenotype resulted from impaired Hh shedding.

**Fig. 3 | Inhibition of endogenous Kuz impairs Hh-dependent eye development. A** The cartoon shows short-range Hh signaling in the *Drosophila* eye disc that later gives rise to the compound eye. Reiterated cell-to-cell Hh signaling from a mobile source moving from posterior (p) to anterior (a) drives photoreceptor (hexagon) differentiation across the eye primordium. The rate of MF movement over time ($t_0 \rightarrow t_{+1}$, black arrows) is Hh-dependent and ultimately determines the number of photoreceptors (ommatidia). The posterior GMR expression domain where endogenous Kuz activity is reduced is shown in pink. **B** Flies expressing GFP served as controls (left). GMR-mediated suppression of endogenous Kuz biofunction severely impaired eye development. Positive control discs, which lack most Hh expression ($hh^{bar3}/hh^{bar3}$), control discs defective in Hh release in clonal disc tissue ($disp^{LacW}$), and discs expressing $^{HA}$Hh proteins in a $hh^{bar3}/hh^{AC}$ background all develop into small eyes. Hh expression in the $hh^{bar3}/hh^{AC}$ background restores eye development. Scale bars: 100 μm. **B'**) Quantification of phenotypes. One-way ANOVA, Sidak's multiple comparison test. ****: $p < 0.0001$, n.s. $= 0.988$, $n = 10$ for all genotypes. Error bars always represent the standard deviations of the means. See Supplementary Table 1 for detailed statistical information. **C** Reduced expression of Ci (expression level in heatmap LUT representation) at the MF (arrowheads) demonstrates reduced Hh release and function in GMR>kuz$^{DN}$ discs. Scale bar: 50 μm. a: anterior, p: posterior. See Supplementary Fig. 2B, C for details.

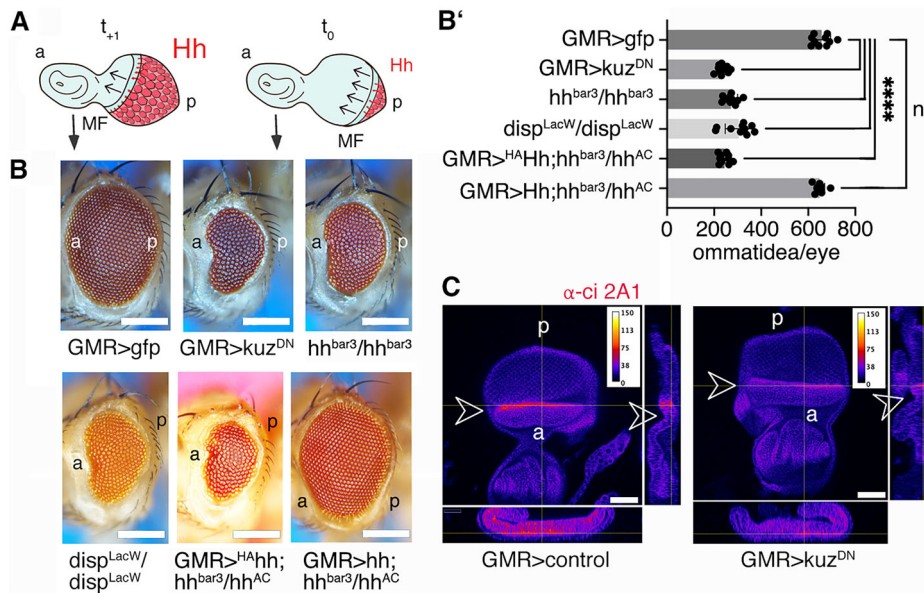

However, we also observed that the Kuz$^{DN}$ phenotype is sensitive to genetic context and is also dependent on the expression level and temperature. Furthermore, the small eye phenotype was often camouflaged by the well-described "rough eye" phenotype as a consequence of impaired Notch signaling upon Kuz$^{DN}$ expression[39]. Therefore, to rule out the potential effects of Notch or other Kuz ligands on the small eye phenotype, we compared the expression of the Hh target protein Cubitus interruptus (Ci) between GMR>kuz$^{DN}$ and GMR>control expressing eye discs. Ci undergoes limited proteolysis in the absence of Hh ligand but converts into an accumulating full-length transcriptional activator in Hh presence. Based on this mechanism, we expected reduced Ci accumulation at the MF in developing eye discs when endogenous Kuz activity and downstream Hh release from adjacent cells are reduced. As shown in Fig. 3C, this is precisely what we observed: robust Ci accumulation occurred in a thin anterior stripe in control eye discs expressing Hh and Kuz endogenously, whereas this peak of Ci accumulation was reduced in eye discs expressing endogenous Hh together with Kuz$^{DN}$, which suppresses endogenous Kuz activity. This result shows that endogenous Hh is released by proteolytic processing in the developing *Drosophila* eye (Supplementary Fig. 2B, C).

### Scube2 enhances A10-mediated processing and release of Shh, but not the general activity of A10

Does Scube2 enhance A10 function specifically or non-specifically? To address this question, we tested the ability of Scube2 to activate proteolytic processing of the established A10 substrate interleukin-2 receptor α (IL-2Rα)[43] (Fig. 4A, A', Supplementary Fig. 3A). We confirmed that shedding of myc-tagged IL-2Rα is A10 dependent, as A10$^{-/-}$ cells released almost no receptor into the media. However, similar IL-2Rα release was observed from control cells regardless of the presence or absence of Scube2 (Fig. 4A, A'). Similar results were obtained for another established A10 substrate, the IL-6Rα (Supplementary Fig. 3B, B')[44,45]. Both results demonstrate that Scube2 is not a general regulator of A10 activity. The specific requirement of Scube2

only for A10-mediated Shh processing (Fig. 1C, C'') provides an unexpected example of how the substrate specificity of an otherwise promiscuous protease[46,47] can be effectively controlled, and prompted us to investigate the underlying mechanism.

### Scube2 EGF domains and the spacer domain are sufficient to increase Shh release

Which Scube2 domains mediate A10-specific Shh release? To address this question, we co-expressed Shh together with Scube2 or two artificial Scube2 variants: Scube2ΔCUB, which lacks the C-terminal CRD and CUB domains, and Scube2ΔEGF, which lacks the nine N-terminal EGF domains (schematics are shown in Fig. 5A). We then compared their ability to enhance Shh solubilization. The positive control Scube2 significantly enhanced A10-driven Shh solubilization (Fig. 5B, B', Supplementary Fig. 4A), confirming our previous observations. Notably, the EGF domains and spacer region of Scube2 (Scube2ΔCUB) enhanced Shh release as well, albeit to a lesser extent than Scube2 (Fig. 5C, C', Supplementary Fig. 4B), and Scube2ΔEGF, which lacks all nine EGF domains, no longer enhanced A10-driven Shh release (Fig. 5D, D', Supplementary Fig. 4C). These results suggest that the Scube2 EGF domains contribute to Shh solubilization.

### Scube2 binds several serum proteins

Next, we asked how EGF domains mediate Shh release. In general, EGF domains are involved in homophilic or heterophilic protein-protein interactions[19,20], such as between the EGF domains of thrombomodulin and the protease thrombin[48], between the EGF domains of the low-density lipoprotein receptor (LDLR) and protein convertase subtilisin[49], or between LDLR-related protein 1 (LRP1) and apolipoprotein E (ApoE) and α-2-macroglobulin[50]. Therefore, we hypothesized that Scube2 may also interact with proteases, protease regulators, or LPPs. To test this hypothesis, we expressed Scube2 in serum-containing medium, bound FLAG-tagged Scube2 to anti-FLAG-tagged Sepharose, and precipitated Scube2 and

**Fig. 4 | Scube2 is not a general enhancer of A10 activity. A** HEK cells and A10$^{-/-}$ cells were transfected with the established A10 target IL-2Rα and A10-mediated proteolytic release was analyzed using antibodies directed against the myc-tagged receptor, both in the presence and absence of Scube2. **A')** Quantification of IL-2Rα release as shown in (**A**). One-way ANOVA, Dunnett's multiple comparison test. ns: $p > 0.05$, ****: $p < 0.0001$, $n = 4–6$. Error bars always represent the standard deviations of the means. See Supplementary Fig. 3 for loading controls and Supplementary Table 1 for detailed statistical information.

interacting proteins. Scube2 interactors were identified by mass spectrometry using a bovine serum database[51] (Table 1, Data are available via ProteomeXchange with identifier PXD071371). This approach identified 11 Scube2 bait peptides along with bovine α-1-antiprotease and α-2-antiplasmin, both serine protease inhibitors. Scube2 also bound fetuin-B and to α-2-macroglobulin, the protein that also binds LRP1, and plasminogen. We note that A10 was not detected because it is not present in serum. Taken together, these results support a role of Scube2 in the indirect regulation of Shh processing at the cell surface through the recruitment of proteases and protease regulators.

Importantly, another Scube2 interactor identified in our screen was apolipoprotein A1 (ApoA1). ApoA1 is the non-mobile signature protein of HDL, which is present in both, serum and interstitial fluid that leaks from the blood capillaries and fills the spaces around peripheral cells[28,52]. We focused on the interaction of Scube2 with HDL for two reasons: First, Hh solubilization requires LPPs, including HDL, both in vivo and in vitro[28,29]. The second reason is that we have recently confirmed that Disp transfers Shh to HDL[27], suggesting a functional relevance of the Scube2 interaction with ApoA1, as detected by mass spectrometric analysis.

### Scube2 control of Shh release becomes increasingly dispensable as extracellular HDL levels rise

To confirm the Scube2 interaction with ApoA1 and HDL, we expressed Scube2, Scube2Δ, and Scube2ΔCUB, bound the FLAG-tagged proteins to anti-FLAG-tagged Sepharose, precipitated interacting serum proteins, and analyzed the precipitate by SDS-PAGE and immunoblotting. We found purified ApoA1 was enriched in Scube2 and Scube2ΔCUB precipitates, but not when Scube2ΔEGF was used (Fig. 6A). The same result was obtained when serum HDL was used (Fig. 6A'). These results confirmed our mass spectrometry findings and fit the observation that Scube2 and Scube2ΔCUB, but not Scube2ΔEGF, strongly increase Shh release in vitro (see Fig. 5). This led us to hypothesize that the interaction between Scube2 and HDL somehow facilitates the Shh transfer from Disp to HDL. To test this hypothesis, we solubilized Shh from HEK cells and A10$^{-/-}$ cells at high extracellular HDL levels (40 μg/ml, which represents about 30–50% of the estimated HDL concentration in human lymph[53]) and in the presence or absence of Scube2. As shown in Fig. 6B and Supplementary Fig. 5A, Shh release in the HDL-containing medium remained A10-dependent (Fig. 6B'), but notably, was rendered completely independent of Scube2 (Fig. 6B"). In addition, the majority of solubilized proteins were N-truncated (Fig. 6B, arrowhead, Supplementary Fig. 5A) and lacked the N-terminal palmitate (Fig. 6C, the soluble HDL-associated protein elutes in fractions #33-35, only a small fraction of dually processed protein elutes in fraction #29, dual-lipidated control R&D Shh elutes in fraction #37, see also ref. 27 for additional information on the methodology). In contrast, unlike Shh released under serum-depleted conditions, HDL-released Shh was not C-terminally processed, suggesting that the cholesterol moiety associates Shh with the soluble carrier[27] and that HDL association protects the C-peptide from A10 cleavage. We note that the HDL-associated N-truncated ligand is bioactive (Supplementary Fig. 5B, C, see also ref. 27 for a thorough characterization of the HDL-associated Shh biofunction) and significantly more active than dual-lipidated Shh that was artificially linked

with HDL before being added to the medium (Supplementary Fig. 5D-D'). These findings show that HDL at high concentrations readily associates Shh, which then signals to target cells despite the lack of the N-palmitate. Again, A10-independent solubilization of artificially monolipidated Shh variants $^{C25A}$Shh and ShhN confirmed that controlled Shh transfer to HDL strictly requires dual lipidation of the PM-associated proteins (Supplementary Fig. 5E, F).

Next, we characterized HDL-driven Shh solubilization in greater detail. As previously observed[27], Shh solubilization became less Scube2 dependent with increasing amounts of HDL (Fig. 6D, Supplementary Fig. 6A). This showed that Scube2 is only required when extracellular HDL availability is low. We confirmed this finding in our A10 expressing HEK293 cells: Scube2 increased Shh truncation and release until HDL concentrations above 20 μg/ml rendered Shh release Scube2-independent. We also made this observation when Scube2ΔCUB was used instead of Scube2 (Fig. 6E, Supplementary Fig. 6B): Scube2ΔCUB increased Shh solubilization up to 20 μg/ml HDL in the medium, but 30 μg/ml or more released high levels of Shh in a Scube2ΔCUB-independent manner. In contrast, Scube2ΔEGF was inactive (Supplementary Fig. 6C). Taken together, these results show that Scube2 binds soluble HDL via its EGF domains to facilitate Disp- and A10-mediated Shh transfer to the LPP, but only when HDL concentrations in the media were low. Of note, and consistent with our results, all nine EGF domains of Scube2 share high sequence identity with the EGF domains of the seven members of the structurally related LDLR family (Table 2). This family includes LDLR, VLDLR, LRP1, LRP1b, LRP2 (also called megalin), LRP4, and apoE receptor-2[54]. While LDLR and VLDLR bind and endocytose ApoB- and ApoE-containing LPPs, respectively, LRP1 is a receptor responsible for cellular uptake of more than 30 macromolecules including ApoE-containing LPPs[55] and proteases or protease inhibitor complexes[56]. LRP2 is an endocytic receptor that also binds and internalizes more than 75 putative ligands, including proteases or protease inhibitor complexes and LPPs. These shared multiligand binding properties between Scube2 and the LDLR family suggest not only structural similarities between the two families of molecules, but also functional similarities. A notable difference between Scube2 and the LDLR family is that ligand binding to the latter results in their internalization and clearance, whereas Scube2 is a soluble molecule and acts in the opposite direction to release Shh from the cell surface.

### HS recruits Scube2, but not ApoA1, to an artificial cell surface

What is a possible explanation for this notable difference? It is known that, following its secretion, Hhs form oligomers at the PM that further concentrate in microscopically visible clusters colocalized with HS[3]. Importantly, clustered basic amino acids in the Scube2 spacer domain also associate the soluble molecule with cellular HS[9,17]. Fluorescence activated cell sorting (FACS) confirms these findings (Supplementary Fig. 7A and[17]). In notable contrast, soluble ApoA1-containing HDL does not bind to HS or heparin[57], and therefore does not readily associate with HS-rich Shh release sites on the cell surface. Based on these considerations, we hypothesized that soluble Scube2 recruits low-abundance HDL to HS-rich Shh release sites, thereby bridging the gap between soluble HDL carriers and PM-associated Disp, A10, and Shh. We tested this hypothesis by measuring the interactions between Scube2, ApoA, or HDL and a reconstituted heparin-decorated PM

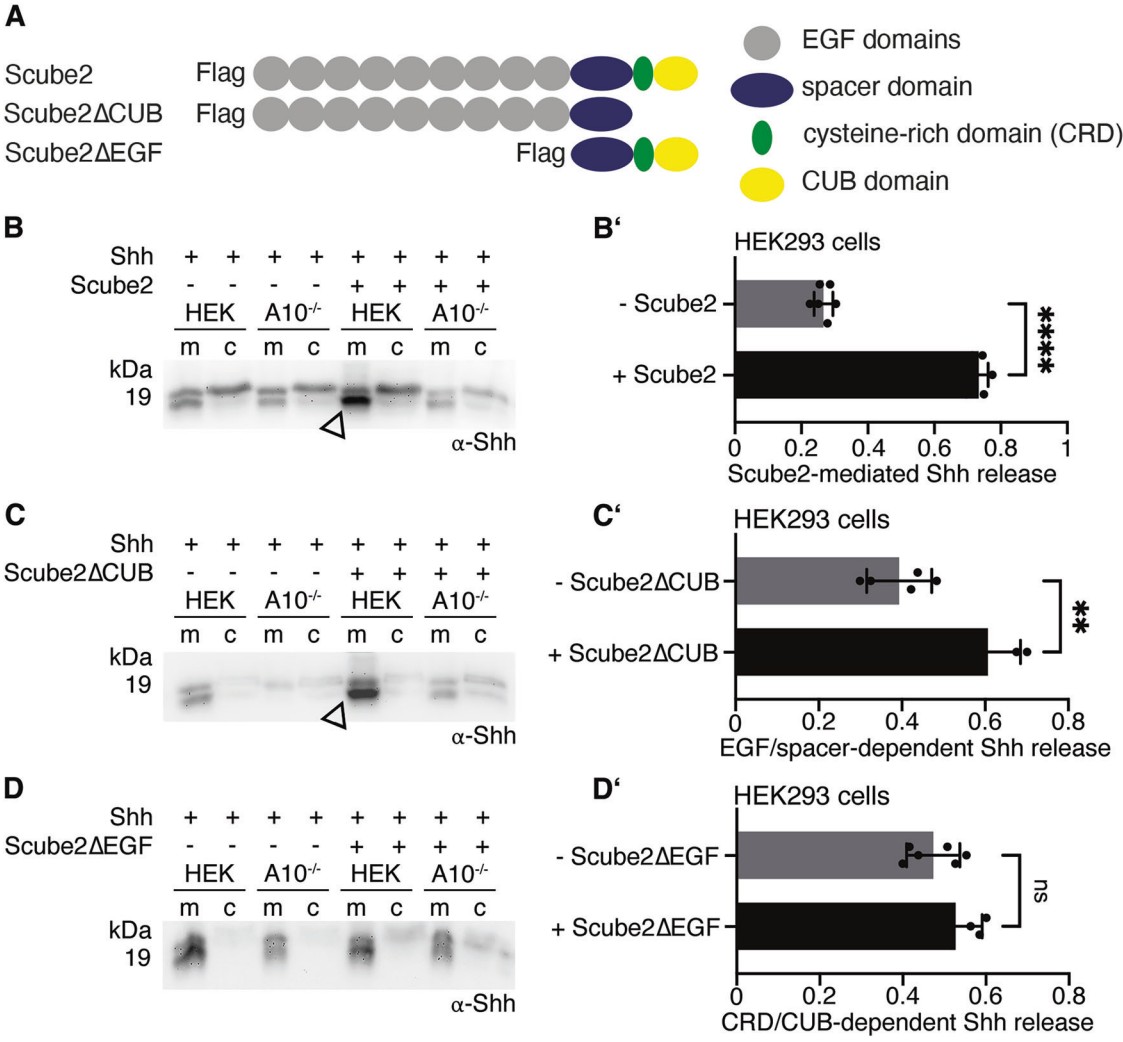

**Fig. 5 | The EGF and spacer domains of Scube2 increase A10-controlled Shh release. A** Schematic of the Flag-tagged Scube2 variants: Scube2ΔCUB consists of all nine EGF domains and the spacer region, while Scube2ΔEGF consists only of the spacer domain, the CRD domain and the CUB domain. **B–D** Shh release from HEK cells or A10⁻ᐟ⁻ cells in the presence or absence of Scube2 or the Scube2 variants. **B** Full-length Scube2 increases A10-mediated Shh solubilization from HEK cells (arrowhead). **C** Scube2ΔCUB also increases Shh solubilization (arrowhead). **D** Scube2ΔEGF does not increase Shh solubilization. **B'-D'** Quantification of Shh solubilization. Unpaired t-test, two-tailed. ****: $p < 0.0001$, **: $p = 0.0026$, ns: $p > 0.05$, $n = 6$ (**B'**), $n = 5$ (**C'**), $n = 6$ (**D'**). Error bars always represent the standard deviations of the means. See Supplementary Table 1 for detailed statistical information.

(heparin is a highly sulfated form of HS) using quartz crystal microbalance with dissipation monitoring (QCM-D). The core of the QCM technology is an oscillating quartz crystal sensor disc with a resonance frequency related to the mass of the disc. This allows real-time detection of nanoscale mass changes on the sensor surface by monitoring changes in the resonance frequency ($\Delta F$): adsorption of molecules on the surface decreases $F$, while a mass decrease increases $F$. QCM-D measures an additional parameter, the change in energy dissipation $D$, where $\Delta D$ tends to increase as mass is added to the sensor surface. As shown in Fig. 6F and Supplementary Fig. 7B,C, incubation of the heparin-functionalized QCM-D chip with purified Scube2 induced a rapid downwards shift in $\Delta F$ of about –20 Hz, confirming HS binding of Scube2 (indicated by an asterisk), and an increased energy dissipation ($\Delta D$). Both parameters were only slightly reversed after washing buffer injection into the QCM-D chambers, indicating limited Scube2 washout from the heparin. Importantly, the addition of soluble heparin to the buffer slightly decreased $F$ and increased $D$. This shows that, unlike Shh that immediately detaches from the immobilized heparin to soluble heparin[58], Scube2 interacts with soluble heparin but remains firmly bound to the immobilized heparin. The conclusion from this experiment is that

soluble Scube2 is effectively recruited to HS-rich surfaces. However, once it is bound, it is unlikely to easily move away from these sites. Importantly, when purified Scube2 was preincubated with purified ApoA1 (Fig. 6G), which alone shows negligible HS interactions (black line), $\Delta F$ decrease and $\Delta D$ increase further due to the added ApoA1 mass on the sensor (arrowhead). This result suggests that Scube2 can target ApoA1 to HS-rich surfaces, which it would otherwise not bind to. We observed similar behavior when Scube2 was preincubated with HDL, although the result was less clear due to a significant portion of HDL interacting with the heparin-functionalized sensor surface on its own (black line). This interaction could be caused by ApoE which can associate with HDL and binds to HS. Nevertheless, HDL alone did not increase $D$, suggesting that the increase in $\Delta D$ in HDL presence is due to the additional HDL mass recruited by Scube2. Taken together, these experiments imply that Scube2 binds to and directs soluble HDL to the HS-rich Shh release sites. In contrast, when HDL concentrations in the media are high, Scube2 recruitment of HDL is less critical due to the law of mass action, which states that the rate of a reaction (here Disp- and A10-mediated Hh transfer to HDL) is directly proportional to the concentrations of the reactants.

**Table 1 | Scube2 interacting serum proteins, as determined by Scube2 immunoprecipitation and mass spectrometric analysis of the associated proteins**

| Protein ID | Protein names | detected Peptides | Coverage [%] | MS/MS count | Function |
|---|---|---|---|---|---|
| Q9NQ36 | Scube2 | 11 | 13.7 | 13 | bait |
| Q7SIH1 | α2-macroglobulin | 14 | 9.7 | 13 | inhibitor of all four proteinase classes |
| Q2UVX4 | complement C3 | 11 | 6.2 | 10 | protease with CUB domain |
| P34955 | α1-antiproteinase | 6 | 17.5 | 26 | proteinase inhibitor |
| Q58D62 | fetuin-B | 4 | 9.6 | 7 | protease inhibitor |
| P06868 | plasminogen | 4 | 4.8 | 3 | protease |
| P15497 | Apolipoprotein A1 | 3 | 11.3 | 3 | HDL constituent |
| P28800 | α2-antiplasmin | 2 | 3.9 | 5 | protease inhibitor |

Albumin was the most prominent hit, but was considered a contaminant and is therefore not listed.

## Discussion

Cell surface Hh release clusters are visible under a light microscope[3]. These clusters also form on HEK293 cells irrespective of Disp expression (Fig. 7A–C), which supports the physiological relevance of our experimental system. Here, we report that A10 contributes to Shh solubilization from these sites, and emphasize this finding through the expression of a dominant-negative form of the *Drosophila* A10 ortholog, Kuz[DN], in the developing eye disc. Furthermore, both this study and a previous one[27] revealed the necessity of HDL at Shh release sites (Fig. 7C), which aligns with the established function of serum LPPs as soluble Hh carriers in vitro and in vivo[28,29]. Moreover, we also showed that A10 specifically cleaves the N-terminal palmitoylated membrane anchor, while the Disp SSD-domain removes the C-terminal cholesteroylated Shh peptide from the cell surface (Fig. 7C–E and[27]). Importantly, we demonstrate that the N-truncated HDL-associated Shh variant released by Disp, Scube2, and A10 is highly bioactive. Thus, Shh's association with HDL not only serves the transport of Hh, but also increases the bioactivity of depalmitoylated Shh, possibly by multimerizing the ligand.

We acknowledge that these results contradict the current Shh release and transport model. This model states that N-palmitate contributes to Shh transport and signaling to the Ptch receptor and must therefore not be removed. This model is supported by a cryo-EM structure of Ptch with the Shh N-terminal palmitate inserted into the cholesterol transport tunnel[34]. However, this structure cannot definitively disprove our data because the Ptch receptor was artificially reconstituted using detergent-extracted, dual-lipidated Shh ligand. The central question arising from this approach is: Are Shh ligands solubilized under physiological conditions (i.e., by Disp and Scube2) also dual-lipidated? Thus far, no direct evidence demonstrating the dual-lipidation of soluble Hh or Shh has been published. Furthermore, alternative explanations for the altered activities of engineered Hh/Shh ligands in vivo have never been considered. A caveat of all studies based on ligands that lack N-palmitate, C-cholesterol, or both during biosynthesis is that monolipidated ligands are immediately released from the cell surface in a Disp- and Scube2-independent, unregulated manner, as demonstrated in this and our previous study[27]. This affects their biofunctions as well and explains why signaling of engineered ligands can increase rather than decrease in some developmental contexts and tissues[59,60]. Therefore, our data support the in vivo relevance of terminal Hh lipids but with an alternative explanation. Finally, we do not see a necessary clash between our data and the current model simply because more than one Hh transport and signaling mode likely exists. Indeed, alternative cryo-EM Shh/Ptch structures are consistent with Ptch activity regulation by Shh independent of the N-terminal palmitoylated peptide[61,62], as observed in this study.

The HDL transport model raised an important question: How can the essential role of Scube2 in facilitating Shh release and signaling be mechanistically explained if it is not the acceptor and transporter for Shh? One clue to answering this question comes from the observation that adding high levels of purified HDL to Shh-expressing HEK293 cells makes Disp- and A10-mediated Shh release Scube2-independent[27] (Fig. 7D). Low HDL levels, in contrast, require their association with the EGF domains of Scube2 to increase Shh release at the PM (Fig. 7E). However, Scube2 is a soluble glycoprotein, which raises another question of how Scube2 increases the amounts of HDL at the sites of Shh release.

Here, the answer comes from published results showing that the formation of punctate PM sites of Hh release (Fig. 7A, B) requires direct interactions between two positively charged polybasic motifs on Hhs[58,63] and negatively charged cell surface HS[3,9]. Disruption of these interactions impairs cluster formation and Hh signaling in vivo[3], which identifies HS as yet another relevant component of the Hh release machinery. Importantly, Scube1 and Scube2 also carry polybasic HS-binding motifs in their spacer domains[17,21] (Fig. 7E) that contribute strongly to Shh release[64]. In contrast, the HDL acceptor for Shh is the only known LPP that does not bind HS/heparin independently[57], unlike the other major LPPs VLDL, LDL, and chylomicrons[53]. This property is exploited in clinical analyses of blood cholesterol where the non-HDL fraction is precipitated with heparin and the remaining soluble HDL is measured[65]. These findings suggest that Scube2 compensates for HDL's inability to bind cell surface HS by acting as a linker between HDL and cell surface HS[17], thereby enriching low-abundance HDL and enabling the law of mass action at the spatially defined Shh release complex (Fig. 7E). Hence, we conclude that the physiological role of Scube2 is to ensure that HDL acceptor levels at Hh release sites are always at 100% capacity and not Shh release-limiting, regardless of their interstitial levels.

Notably, the facilitator function of Scube2 under conditions of low HDL abundance—as opposed to its proposed role as an essential Shh carrier itself[14,15] – is supported by targeted Scube2 knockouts in mice and zebrafish, which result in relatively mild and tissue-restricted developmental defects[10,66]. Furthermore, a recent publication demonstrated the importance of EGF domains for Scube2 function in vivo[67]. These results raise a final question: Why does *Drosophila melanogaster* not express a Scube2 ortholog, despite the otherwise complete conservation of the Hh release machinery? Importantly, this machinery also includes LPPs—known as lipophorins in insects—that play an essential role in Hh release and transport in vivo[28,29]. We suggest that one possible explanation for the absence of Scube orthologs in *Drosophila*, despite the conserved role of LPPs in Hh release and transport[28,29], is that insects have an open circulatory system in both larvae and adults. Here, the hemocoel directly bathes developing cells with high levels of lipophorins without distinguishing between "blood" and "interstitial fluid". Therefore, lipophorin concentrations in the hemocoel may be high enough to readily act as effective Hh acceptors, similar to the "high HDL" conditions used in our cell culture experiments that rendered Scube2 function unnecessary. Another difference between vertebrate and insect LPPs is that lipophorin, unlike HDL, binds HS on the cell surface[68] and therefore may not require a facilitator protein to reach Hh/HS clusters at the PM. A final possibility to explain the lack of a Scube ortholog in insects is that the soluble protein

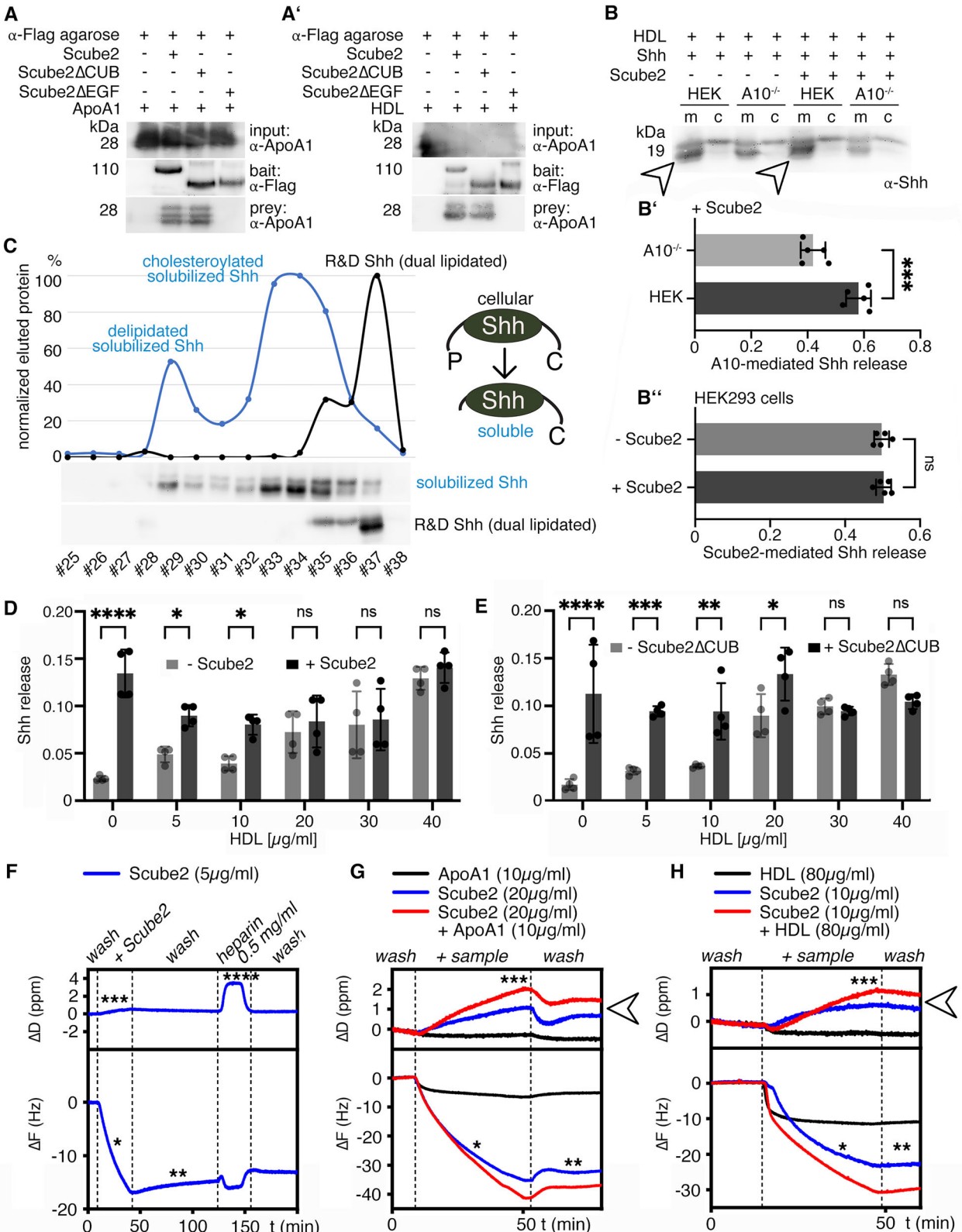

Shifted (Shf, which has five EGF-like domains and affects Hh signaling in vivo) may perform this function instead[69]: Like Scube2 in vertebrates, Shf in *Drosophila* acts over long distances, interacts with HS on Hh source cells, and acts only on cholesteroylated Hh in a manner depending on the EGF domains[70]. Therefore, it will be interesting to see whether Shf also interacts with the insect LPP, lipophorin.

## Materials and methods

### Fly lines

The following fly lines were used: GMR-Gal4 (GMR > ): GMR17G12 (GMR45433-GAL4):            P(y[+t7.7]w[+mC]=GMR17G12-GAL4)attP2, Bloomington stock #45433 (discontinued but available from our lab), w; P(w[+mC]=UAS-kuz.DN)2, Bloomington stock #6578, and flies

**Fig. 6 | Scube2 and a minimal EGF domain/spacer construct bind HDL and HS.**
**A** Purified ApoA1 co-immunoprecipitates with the Scube2 bait and also with the
Scube2ΔCUB bait, but not with Scube2ΔEGF. **A'**) HDL from serum co-
immunoprecipitates with the Scube2 bait and the Scube2ΔCUB bait. **B** 40 µg/ml
purified human HDL in serum-free media solubilized Shh independently of Scube2.
Soluble truncated Shh is indicated by arrowheads. **B'**) Quantification of Shh solu-
bilization from HEK cells or A10$^{-/-}$ cells, as shown in (**B**). Unpaired $t$ test, two-tailed.
****: $p = 0.0004$, $n = 5$ independent assays. **B''**) Quantification of Shh solubilization
from HEK control cells, as shown in (**B**). Unpaired $t$ test, two-tailed. ns: $p = 0.69$.
n = 5 independent assays. **C** RP-HPLC confirmed the loss of the N-terminal pal-
mitoylated Shh peptide during Disp-, A10- and Scube2-regulated solubilization and
showed that the C-terminal cholesterol (C) is sufficient to associate the N-processed
Shh with HDL[27]. **D** Shh solubilization in medium containing increasing con-
centrations of HDL in the presence or absence of Scube2. Two-way ANOVA. ****:

$p < 0.0001$, *: $p < 0.05$, ns: $p > 0.05$, $n = 4$. **E** Scube2ΔCUB and Scube2 have similar
Shh release-enhancing activity profiles at low HDL levels. Two-way ANOVA. ****:
$p < 0.0001$, ***: $p < 0.001$, **: $p < 0.01$, *: $p < 0.05$, ns: $p > 0.05$, $n = 4$. Error bars
always represent the standard deviations of the means. **F** Purified Scube2 (blue)
binds to an HS-functionalized artificial PM, as indicated by a decrease in $F$ (one
asterisk) and a slight increase in $D$ (three asterisks) during Scube2 binding in QCM-
D measurements. The protein remains associated with the surface during an
extensive buffer wash (two asterisks) and when soluble heparin is added to the buffer
(four asterisks). **G, H** Scube2 pre-incubated with ApoA1 (G, red line) or HDL (H, red
line) is also recruited to the sensor surface (asterisk) and remains bound during
washing (two asterisks), with an additional decrease in $F$. The additional increase in
$D$ (blue vs. red lines, arrowheads) suggests the recruitment of ApoA1 and HDL by
Scube2 to the functionalized surface. See Supplementary Fig. 7 for detailed infor-
mation on the QCM-D technique.

**Table 2 | Scube2 EGF domains show high sequence identity with the EGF consensus sequences of the LDLR and the LRPs**

|  | LDLR | LRP1 | LRP2 | LRP4 | LRP5 | LRP6 | LRP8 |
|---|---|---|---|---|---|---|---|
| EGF1 | 50.00% | 35.29% | 39.39% | 35.29% | 32.26% | 30.00% | 31.43% |
| EGF2 | 52.00% | 38.71% | 42.86% | 46.88% | 43.75% | 50.00% | 48.00% |
| EGF3 | 55.17% | 41.67% | 41.67% | 44.44% | 44.74% | 41.67% | 52.78% |
| EGF4 | 37.14% | 37.84% | 37.84% | 35.14% | 35.14% | 40.54% | 40.54% |
| EGF5 | 38.24% | 46.67% | 50.00% | 38.89% | 54.17% | 33.33% | 41.67% |
| EGF6 | 51.72% | 51.53% | 47.22% | 47.22% | 47.22% | 50.00% | 55.56% |
| EGF7 | 35.29% | 48.39% | 45.16% | 44.44% | 47.22% | 41.67% | 47.22% |
| EGF8 | 35.71% | 46.43% | 35.29% | 45.83% | 40.91% | 40.91% | 33.33% |
| EGF9 | 48.28% | 45.19% | 43.33% | 44.44% | 36.11% | 42.11% | 40.00% |
| combined | 42.50% | 22.11% | 43.02% | 42.50% | 44.74% | 44.44% | 42.11% |

This indicates similar functions of Scube2 EGF-domains and the EGF-domains of the LDLR and of LRP1-6.

homozygous for *UAS-hh* or *UAS-*$^{HA}$*hh*[35]. We used an established protocol[58]
that was adapted to delete disp function from large clones in the eye disc. In
brief, the hsFLP in flies carrying a disp hypomorphic disp$^{S037707}$ allele[23]
flanked by an FRT site (P(ry[+t7.2]=hsFLP)12,y[1]w[*];P(ry[+t7.2]
=neoFRT)82B;P(w[+mC]=lacW)disp[S037707]/TM6B,Tb[1], Bloo-
mington stock #53711) was replaced with eye disc specific eyFlp3.5 and an
Rps3 allele to generate large mutant clones, generating flies with the geno-
type eyFlp3.5/ + ;FRT82 P(w[+mC]=lacW)disp[S037707]/ FRT82 Rps3
UbiGFP for Fig. 3B. Kuz$^{DN}$ transgene expression in the morphogenetic
furrow of the eye disc was conducted by crossing both lines at 25°C.
Ommatidia number of the resulting *UAS-Kuz$^{DN}$/GMR-Gal4* flies were
analyzed with a Nikon SMZ25 microscope. GMR>gfp flies served as positive
controls and +/+;hh$^{bar3}$/hh$^{bar3}$ flies served as negative controls. y[1] w[*];
P(w[+m*]=GAL4-ey.H)3-8, P(w[+mC]=UAS-FLP.D)JD1; P(ry[+t7.2]
=neoB)82B P(w[+mC]=GMR-hid)SS4, l(3)CL-R[1]/TM2 (Bloomington
stock# 5253) served as positive controls in apoptosis assay. These flies
express head involution defective (hid), an activator of apoptosis, under
GMR control in the eye disc. To analyze Cubitus interruptus expression in
the eye disc, we crossed flies carrying recombinant GMR-Gal4 UAS-mCD8-
GFP with UAS-kuz$^{DN}$ or UAS-FLP-JD1 (BL#4539) to control for Gal4
levels. The flies were reared at 25 °C and wandering third instar larvae
collected for immunofluorescence analysis. All specimens for quantification
were processed with the same batches of diluted primary and secondary
antibodies in 10% goat serum and 0.3% Triton X-100 in PBT. They were
mounted individually in VectaShield with #1 coverslips as spacers to stan-
dardize the Z dimension. The following antibodies were used: α-GFP
(rabbit, Thermo Fisher Scientific, A-6455) at a dilution of 1:1000, goat-α-
rabbit 488 (Dianova, #115-545-071) at a dilution of 1:500, and goat-α-rat
568 (Dianova, 112-165-003) at a dilution of 1:500. The 2A1 α-Ci (DSHB)
antibody to detect Hh signaling activity at the morphogenetic furrow in
*Drosophila* eye discs was obtained from the Developmental Studies

Hybridoma Bank, created by the NICHD of the NIH and maintained at the
University of Iowa, Department of Biology, Iowa City, IA 52242, and was
used at a 1:200 dilution. DAPI was diluted 1:10,000. All image data were
acquired with identical laser and detector settings at a Zeiss LSM 880.
Acquisition in Z was standardized to 25 equidistant optical sections in all
specimens. All image data were processed identically: LSM image stacks
were imported, orthogonal sections were processed, the 8-bit histogram
range was thresholded down from 255 to 150 to enhance contrast, and a heat
map lookup table (FIRE, Fiji) was applied. For the data shown in Supple-
mentary Fig. 2B, we analyzed seven control imaginal discs from five indi-
viduals and eight imaginal discs from four GMR>kuz$^{DN}$ larvae. We
calculated the mean of the histogram gray-level pixel sums (8-bit, 256 bins,
1024 × 1024 x 25 = 26,214,400 pixels per sample) and plotted them with a
log scale on the y-axis. The error bars indicate the standard deviation. From
the same dataset, we plotted the pixel count ratio for each gray level and
shortened the x-axis to show only the relevant gray levels (see Supple-
mentary Fig. 2B, bottom). The incremental thresholding image series shown
in Supplementary Fig. 2C was created based on a maximum intensity
projection with 8 bits (256 levels of gray). Pixel levels 0-20 are shown in blue,
pixel levels X-255 are shown in green, and grayscale pixel levels 20-X are
shown, with X as indicated in Supplementary Fig. 2C.

## Cell lines
The generation and validation of Disp knockout cells (Disp$^{-/-}$) and HEK
control cells was previously described[22]. To specify the ADAM target regions
for the CRISPR/Cas9-mediated genome editing, the third exon of the
human A10 genomic sequence (ENST00000260408) and the first exon of
the human A17 genomic sequence (ENST00000310823) were submitted to
an online CRISPR Design Tool (http://tools.genome-engineering.org). For
each targeted gene, a pair of oligonucleotides (A10g#2_F: 5'ACCGA-
TACCTCTCATATTTACAC, A10g#2_R: 5'AACGTGTAAATATGAGA

**Fig. 7 | Model of Disp-mediated Shh transfer to HDL.** Disp-mediated Shh solubilization requires HDL and A10, whereas the requirement for Scube2 is relative and depends on the amount of available HDL. TIRF-SIM of Shh-transfected HEK293 cells (**A**) or Disp[−/−] cells (**B**) using the monoclonal anti-Shh antibody 5E1 shows unchanged cell surface clustering of overexpressed Shh in both cell lines. Scale bar: 10 μm. **C** Dual-lipidated Shh colocalizes with HSPGs[3] (left). The availability of HDL carriers at the cell surface is a critical step for Shh release: In the absence of HDL, the Disp-regulated release complex is incomplete and Shh cannot be released, even in the presence of Scube2[27]. **D** High HDL levels result in frequent random encounters of HDL with the Shh release complex based on the law of mass action. This allows Disp to transfer the C-terminal Shh cholesterol moieties to HDL. This first step of Shh extraction exposes the N-terminal cleavage site for subsequent proteolytic processing by A10. Both activities release Shh from the PM. **E** We propose that one role of Scube2 is to increase the frequency of HDL encounters with the release complex when HDL levels are low. This may be achieved by recruiting the soluble HDL/Scube2 complex to the punctate Shh/HS clusters at the cell surface[3,9,17]. See Discussion for details.

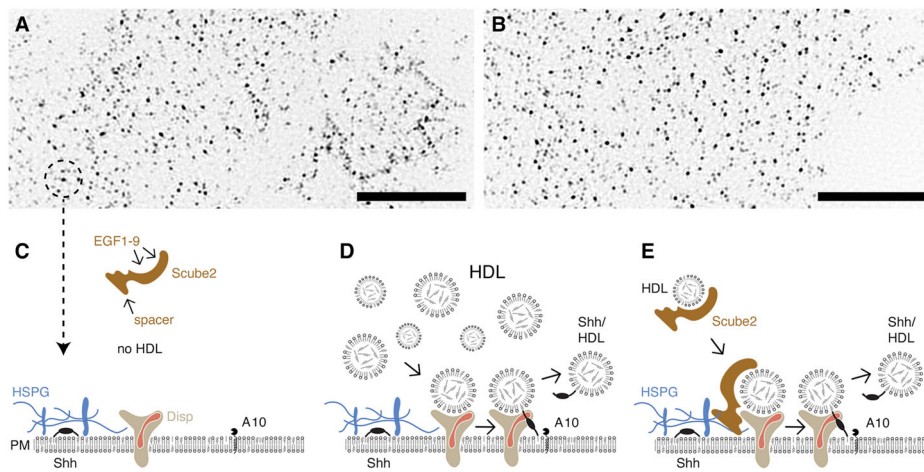

GGTATC; A17g#2_F: 5'ACCGCCGCGACCTCCGGATGACC, A17g#2_R: 5'AACGGTCATCCGGAGGTCGCGGC) was annealed, phosphorylated and subsequently cloned into the SapI-digested guideRNA expressing plasmid LeGO-Cas9-iC-puro+cgRNA-SapI (generously provided by Boris Fehse, UKE Hamburg). The resulting plasmids were designated LeGO-Cas9-A10 and LeGO-Cas9-A17, respectively. Genome-editing to achieve protease-deficient HEK293 cells – CRISPR/Cas9 plasmids were transfected into HEK293T cells using TurboFect Transfection Reagent (Life Technologies, Carlsbad, CA). 48 h following transfection, cells were grown in the presence of 1 μg/ml puromycin for additional 48 h to enrich successfully transfected cells. For the isolation of genome-edited cells, cell populations were subjected to FACS (fluorescence activated cell sorting) analysis after immunostaining with PE anti-human CD156c antibody (BioLegend, San Diego, CA) and A300E antibody (Institute of Biochemistry, Kiel, Germany) and single-cell sorted into 96-well plates. Single cell clones were further expanded and the efficient knock-out of A10 and A17 confirmed via FACS and Western blotting. Complete A10 knock-out was finally confirmed by sequencing. Importantly, Disp and A10/A17 were deleted in HEK293 cells or the HEK293 cell derivative Bosc23, which makes our results comparable to earlier assays that were conducted in this cell type[14,15]. Disp[−/−], HEK, A10[−/−], and A17[−/−] cells, and C3H10T1/2 reporter cells were maintained in DMEM supplemented with 10% FCS and 100 μg/ml penicillin-streptomycin. Sequencing of A10[−/−] cells revealed deletions in 3 different A10 loci, consistent with the modal chromosome number of 64 in HEK293 cell lines (https://www.phe-culturecollections.org.uk/products/celllines/) that introduced stop codons at or upstream of codon#110 in the inhibitory N-terminal propeptide sequence. All cells tested negative for mycoplasma contamination.

### Cloning of recombinant proteins

Shh expression constructs were generated from murine cDNA (NM_009170: nucleotides 1-1314, corresponding to amino acids 1-438; and ShhN: nucleotides 1-594, corresponding to amino acids 1-198) and human Hhat cDNA (NM_018194). Both cDNAs were cloned into pIRES (Clontech) for their coupled expression from bicistronic mRNA to achieve near-quantitative Shh palmitoylation[16]. ShhN (nucleotides 1-594, corresponding to amino acids 1-198) and Hhat were also cloned into pIRES. [C25S]Shh was

generated by site-directed mutagenesis (Stratagene). Unlipidated [C25S]ShhN cDNA and non-palmitoylated [C25S]Shh cDNA (amino acids 1-438) were inserted into pcDNA3.1 (Invitrogen). Primer sequences can be provided upon request. cDNAs encoding IL-2Rα and IL-6Rα were provided by the Garbers lab. Human Scube2 constructs were a kind gift from Ruey-Bing Yang (Academia Sinica, Taiwan). Where indicated, dual-lipidated, HEK293-derived human Shh (R&D Systems, 8908-SH) served as a bioactivity reference and to quantify Bosc23-expressed, TCA-precipitated proteins on the same blots.

### Protein detection

HEK cells, Disp[−/−] cells and A10[−/−] cells were seeded into six-well plates and transfected with 1 μg Shh constructs together with 0.5 μg Scube2 or empty cDNA3.1 using Polyfect (Qiagen, #3011071). Cells were grown for 2 days at 37 °C with 5% CO$_2$ in DMEM containing 10% FCS and penicillin-streptomycin (100 μg/ml). Serum-containing media were aspirated and serum-free DMEM added for 6 h, harvested, and centrifuged at 300 × g for 10 min to remove debris. Supernatants were incubated with 10% trichloroacetic acid (TCA) for 30 min on ice, followed by centrifugation at 13,000 × g for 20 min to precipitate the proteins. Cell lysates and corresponding supernatants were analyzed on the same reducing SDS polyacrylamide gel and detected by Western blot analysis by using rabbit-α-Shh antibodies (Cell signaling C9C5), rabbit-α-GAPDH antibodies (Cell Signaling, GAPDH 14C10, #2118), or α-β-actin antibodies (Sigma-Aldrich, A3854) followed by incubation with horseradish peroxidase-conjugated secondary antibodies. FLAG-tagged Scube2 was detected by using polyclonal α-FLAG antibodies (Sigma, St. Louis, USA). GAPDH, β-actin (for cell lysates), or Ponceau S (for media) served as a loading control. Note that the amounts of immunoblotted soluble and cellular Shh do not correlate inversely. This is because medium lanes represent all TCA-precipitated proteins, while cells were directly lysed in SDS buffer and only a small fraction (about 5%) were applied to the gel. As a consequence, a 100% increase in Shh solubilization will correlate to only 5% reduction in the amount of cell-surface-associated Shh, and vice versa. Band intensities on Western blots were quantified using ImageJ, and the ratio of released soluble protein to the corresponding cellular precursor was calculated as follows: first, the gray-scale ratio between the soluble and cellular proteins was

calculated using the respective quantifications. Then, to correct for different signal strengths on Western blots derived from multiple independent experiments, all ratios derived from the same blot were summed up, and each medium/cell ratio from that blot was divided by the sum. Therefore, to allow for reliable comparison, all samples in individual release experiments required loading onto the same blot. This protocol was varied in three ways: For *serum-free* release, cells cultured in DMEM + 10% FCS were carefully washed three times with serum-free DMEM before serum-free media were added for 6 h of protein release. For Shh release into *serum-depleted* medium, cells were not washed before the serum-free DMEM was added. For release into *HDL-containing* media, cells cultured in DMEM + 10% FCS were carefully washed three times with serum-free DMEM before serum-free media supplemented with up to 40 µg/ ml HDL were added for 12 h of protein release. The longer incubation time was required due to the loss of cells during the washing steps and the corresponding reduction in released proteins per time and culture well. For large-scale expression of Scube2, HEK293 cells were grown to 80–90% confluence in 15 cm dishes and the medium was replaced with 16.6 ml DMEM containing 10% FCS. A premix consisting of 18.4 µg Scube2 and 276 µl polyethyleneimine (PEI) from stock (1 mg/ml in $H_2O$, pH 7.0) in 920 µl medium without growth factors, serum, antibiotics or other proteins was incubated for 8 min. The premix was then supplemented with 8.3 ml DMEM containing 10% serum and added dropwise to the dish. The cells were incubated at 37 °C. 8 h after transfection, the medium was replaced with fresh DMEM containing serum and antibiotics and incubated for a further 24 h. The medium was then aspirated, the cells were washed three times in DMEM without serum or antibiotics, and Scube2 was secreted into 20 ml of DMEM without serum or antibiotics for 24 h. The supernatants were then transferred to 50 ml Falcon tubes, centrifuged for 10 min at 14,000 rpm and the clarified media concentrated by Amicon ultrafiltration using membranes with a 30 kDa cut-off. Scube2 in the concentrates was quantified by SDS-PAGE prior to use in subsequent QCM-D experiments.

## TUNEL assay

Apoptotic cells in L3 eye discs were quantified using the TdT dUTP nick end labeling (TUNEL) assay (Biomol, #E-CK-A325.50). Briefly, eye discs were transferred to 1.5-ml reaction tubes and fixed with 4% paraformaldehyde at room temperature for 20 min. Then, the discs were permeabilized with 0.3% Triton X-100 in PBS at 37 °C for 10 min. The TUNEL reaction was carried out in the dark using 250 µl of reaction mixture at 37 °C for 60 min. The cell nuclei were stained with DAPI, and the discs were immediately examined under an LSM 700 fluorescence microscope (Zeiss).

## TIRF-SIM

nt-Ctrl and Disp$^{-/-}$ cells were seeded on gelatin-coated glass slides and transfected with Shh. Fixation was performed in 4% paraformaldehyde (PFA) and 2% glutaraldehyde for 10 min at room temperature, cells were washed 3 times with PBS and blocked with 5% goat serum in PBS for 1 h. Cells were then incubated overnight with α-Shh antibody (5E1 mouse, 1:250, DSHB, overnight) in blocking buffer at 4 °C. The next day, cells were washed and incubated with FITC-labeled anti-mouse secondary antibody (1:300, Dianova, #111-095-144) for 2 h at room temperature. Analysis was performed on an LSM Airyscan detector using a 63x objective, with data deconvolved using Huygens Professional X11. Maximum intensity projections were generated using Fiji.

## Shh bioactivity assay

Shh-conditioned HDL-containing DMEM was sterile filtered, FCS was added to the fractions at 10% and mixed 1:1 or 1:2 with DMEM supplemented with 10% FCS and 100 µg/ml antibiotics, and the mixture was added to C3H10 T1/2 cells. Where indicated, dual-lipidated, HEK293-detergent extracted human Shh (R&D Systems, 8908-SH) mixed with human HDL served as a bioactivity control. Gel filtration analysis showed that the dually lipidated detergent-solubilized protein associated quickly and spontaneously with the HDL, shifting in size from 20 kDa (monomeric Shh) to

200-400 kDa (Shh associated with HDL, not shown). Cells were harvested 5 days after osteoblast differentiation was induced and lysed in 1% Triton X-100 in PBS, and osteoblast-specific alkaline phosphatase activity was measured at 405 nm by using 120 mM p-nitrophenolphosphate (Sigma) in 0.1 M Tris buffer (pH 8.5). Values measured in mock-treated C3H10 T1/2 cells served as negative controls.

## Quantitative PCR (qPCR)

Alternatively, C3H10T1/2 cells were stimulated with recombinant Shh/ HDL with or without Scube2 in triplicate for 2 days. TriZol reagent (Invitrogen, #1014257) was used for RNA extraction from C3H10T1/2 cells. A first strand DNA synthesis kit and random primers (Thermo, Schwerte, Germany) were used for cDNA synthesis before performing a control PCR with murine β-actin primers. Amplification with Rotor-Gene SYBR-Green on a BioRad CFX 384 machine was conducted in triplicate according to the manufacturer's protocol by using the following primer sequences: actin forward: 5′CTATTGGCAACGAGCGGTTC, actin reverse: 5′ CGGATGTCAACGTCACACTTC, Ptch1 forward: 5′GGGCTACGAC-TATGTCTCTC, Ptch1 reverse: 5′CTTTGATGAACCACCTCCAC, Gli1 forward: 5′CCCTGGTGGCTTTCATCAAC, and Gli1 reverse: 5′TGACT-CATCTGAGGTGGGAATC. Cq values of technical triplicates were averaged, the difference to β-actin mRNA levels calculated by using the ΔΔCt method, and the results expressed as log2-fold change if compared with the internal control of C3H10T1/2 cells stimulated with mock-transfected HDL-containing media.

## Reverse-phase high performance liquid chromatography (RP-HPLC)

HEK293 cells were transfected with expression plasmids for dual-lipidated Shh, unlipidated $^{C25A}$ShhN control protein, cholesteroylated (non-palmitoylated) $^{C25A}$Shh, and palmitoylated ShhN. Where indicated, human Shh (R&D Systems, 8908-SH) served as the dual-lipidated control protein. Two days after transfection, cells were lysed in radioimmunoprecipitation assay buffer containing complete protease inhibitor cocktail (Roche, Basel, Switzerland, #11697498001) on ice and ultracentrifuged, and the soluble whole-cell extract was acetone precipitated. Protein precipitates were resuspended in 35 µL of (1,1,1,3,3,3) hexafluoro-2-propanol and solubilized with 70 µL of 70% formic acid, followed by sonication. RP-HPLC was performed on a C4-300 column (Tosoh, Tokyo, Japan) and an Äkta Basic P900 Protein Purifier. To elute the samples, we used a 0–70% acetonitrile/ water gradient with 0.1% trifluoroacetic acid at room temperature for 30 min. Eluted samples were vacuum dried, resolubilized in reducing sample buffer, and analyzed by SDS-PAGE and immunoblotting. Signals were quantified with ImageJ and normalized to the highest protein amount detected in each run.

## Size exclusion chromatography (SEC) chromatography

Shh size distribution in the presence of HDL was confirmed by SEC analysis with a Superdex200 10/300 GL column (GE Healthcare, Chalfront St. Giles, UK, GE28-9909-44) equilibrated with PBS at 4 °C fast protein liquid chromatography (Äkta Protein Purifier (GE Healthcare)). Eluted fractions were TCA-precipitated, resolved by 15% SDS-PAGE, and immunoblotted. Signals were quantified with ImageJ.

## Scube2 pull down

For FLAG-tagged Scube2 pull down, HEK cells were seeded into six-well plate and transfected with 1 µg Scube2 by using Polyfect (Qiagen). Cells were incubated for 2 days at 37 °C with 5% $CO_2$. For protein pull-downs, anti-FLAG M2-agarose beads (Sigma, F2426) were used. For each pull down, 40 µl beads were added to 1 ml sample, the samples incubated over night at 4 °C on a rotator and centrifuged at 8000 × g for 30 s. After the supernatant was discarded, the precipitates were washed three times using 500 µl TBS. Beads were again centrifuged at 8000 × g for 30 s between the washing steps and the supernatant was discarded. Proteins were then eluted from the beads using 30 µl reducing SDS buffer and were incubated for

10 min at 85 °C. After centrifugation at $8000 \times g$ for 30 s, the supernatants were loaded to a 15% SDS polyacrylamide gel and proteins detected by Western blot analysis using polyclonal α-FLAG antibodies (Sigma, F7425) and α-ApoA1 antibodies (ABIN7427912, antibodies-online.com). HDL-input controls were obtained by protein precipitation and immunodetection of ApoA1 in the sample.

## Scube2 expression and purification

For large-scale expression of Scube2 tagged with an hexahistidine tag at the N-terminus, HEK293 cells were grown to 80–90% confluence in 15 cm dishes and the medium was replaced with 16.6 ml DMEM containing 10% FBS. A premix consisting of 18.4 µg Scube2 and 276 µl polyethyleneimine (PEI) from stock (1 mg/ml in $H_2O$, pH 7.0) in 920 µl medium without growth factors, serum, antibiotics or other proteins was incubated for 8 min. The premix was then supplemented with 8.3 ml DMEM containing 10% serum and added dropwise to the dish. The cells were incubated at 37 °C. 8 h after transfection, the medium was replaced with fresh DMEM containing serum and antibiotics and incubated for a further 24 h. The medium was then aspirated and Scube2 was secreted into 20 ml of DMEM without serum or antibiotics for 24 h. The supernatants were then transferred to 50 ml Falcon tubes, centrifuged for 10 min at 14,000 rpm and the clarified media concentrated by Amicon ultra-filtration using membranes with a 30 kDa cut-off. Next, the concentrated solution was applied to an Äkta System (GE Healthcare) using a 1 ml Protino NiNTA column (Macherey Nagel, #745410.5) and a flowrate of 1 ml/min. The column was washed with buffer (wash buffer A, 10 mM Tris, 100 mM NaCl (Sigma Aldrich) at pH 7.4) for 30 min. The tagged protein was then eluted and fractioned with 250 mM imidazole in wash buffer A for 15 min at a flow rate of 1 ml per min. Fractions 9 and 10 where pooled and samples were taken for SDS-PAGE analysis to determine protein purity and concentration by using BSA standards of known concentration followed by densitometric analysis. The remaining sample was dialyzed, aliquoted, snapfrozen in liquid nitrogen and stored at –80 °C until further use.

## Synthesis of biotinylated heparin

Biotinylated heparin was synthesized by adapting a previously reported procedure[71]. Heparin (4 mM, AppliChem, A3004.0005) was dissolved in 100 mM acetate buffer (made from glacial acetic acid (Carl Roth, Karlsruhe, Germany, #3738.1) and sodium acetate (Sigma-Aldrich) at pH 4.5) containing aniline (100 mM, Sigma-Aldrich, #8.22256.1000). Biotin-PEG₃-oxyamine (3.4 mM, Conju-Probe, San Diego, USA, CP3041) was added to the heparin solution and allowed to react for 48 h at 37 °C. The final product was dialyzed against water for 48 h by using a dialysis membrane with a 3.5 kDa cutoff. The obtained biotinylated heparin was characterized by biotin-streptavidin binding assays using QCM-D. The average mass of the heparin, when anchored to the surface, was estimated at 9 kDa ( ~ 18 disaccharide units) by QCM-D analysis[72].

## Preparation of small unilamellar vesicles (SUVs)

SUVs were prepared by adapting reported procedures[73,74]. A mixture of lipids composed of 1 mg/ml 1,2-dioleoyl-sn-glycero3-phosphocholine (DOPC, Avanti Polar Lipids, A80375) and 5 mol% of 1,2-dioleoyl-sn-glycero-3-phosphoethanolamine-N-(cap biotinyl) (DOPE-biotin, Avanti Polar Lipids, A87273) was prepared in chloroform in a glass vial. Subsequently, the solvent was evaporated with a low nitrogen stream while simultaneously turning the vial in order to obtain a homogenous lipidic film. The residual solvent was removed for 1 h under vacuum. Subsequently, the dried film was rehydrated in ultrapure water to a final concentration of 1 mg/ml and vortexed to ensure the complete solubilization of the lipids. The lipids were sonicated for about 5 min until the opaque solution turned clear. The obtained SUVs were stored in the refrigerator and used within 2 weeks.

## QCM-D measurements

QCM-D measurements were performed with a QSense Analyser (Biolin Scientific, Gothenburg, Sweden) and $SiO_2$-coated sensors (QSX303, Biolin Scientific). The measurements were performed at 22 °C by using four parallel flow chambers and one peristaltic pump (Ismatec, Grevenbroich, Germany) with a flow rate of 75 µl per min. The normalized frequency shifts $\Delta F$, and the dissipation shifts $\Delta D$, were measured at six overtones (i = 3, 5, 7, 9, 11, 13). The fifth overtone (i = 5) is presented in this paper; all other overtones gave qualitatively similar results. QCM-D sensors were first cleaned by immersion in a 2 wt% sodium dodecyl sulfate solution for 30 min and subsequently rinsed with ultrapure water. The sensors were then dried under a nitrogen stream and activated by 10 min treatment with a UV/ozone cleaner (Ossila, Sheffield, UK). For the formation of supported lipid bilayers (SLBs), after obtaining a stable baseline, freshly made SUVs were diluted to a concentration of 0.1 mg/ml in buffer solution (wash buffer A, 10 mM Tris, 100 mM NaCl (Sigma Aldrich) at pH 7.4) containing 10 mM of $CaCl_2$ directly before use and flushed into the chambers. The quality of the SLBs was monitored in situ to ascertain high-quality SLBs were formed, corresponding to equilibrium values of $\Delta F = -24 \pm 1$ Hz and $\Delta D < 0.5 \times 10^{-6}$. Afterward, a solution of streptavidin (150 nM) was passed over the SLBs, followed by the addition of biotinylated heparin (10 µg per ml). Each sample solution was flushed over the QCM-D sensor until the signals equilibrated and subsequently rinsed with wash buffer A (see above). Before the addition of HDL and Scube2, $F$ and $D$ were set to 0 and the flow rate was reduced to 20 µl per min. Analysis was performed using QSoft401 version 2.7.3.883.

## FACS

HEK293-derived Bosc23 cells were transfected with Scube2 and a Scube2 variant lacking the major HS-binding amino acid motif, non-enzymatically removed from the culture dish by using Versene (PAA), and suspended in PBS containing 5% FCS in a total volume of 0.5 ml. Scube2-transfected cells were incubated with heparinases I to III (AMS Biotechnology) at 37 °C or with 10 µg/ml heparin (AppliChem) at 4 °C for 1 h. Cells were washed and treated with α-FLAG antibody (1:500 dilution) for 1 h and fluorescein isothiocyanate-conjugated goat-α-rabbit secondary antibody (1:200 dilution, Dianova) for 30 min on ice. FACS analysis was performed on a BD Accuri C6 flow cytometer (BD Biosciences). Histograms were created by using FlowJo single cell analysis software.

## Mass spectrometry

Scube2 was expressed and precipitated as before using the pull down. Samples were eluted using 0.1 M glycine HCl pH 3.5. 100 µL 0.1 M glycine HCl were added to each sample and incubated for 10 min at room temperature. Afterwards, samples were centrifuged at $8200 \times g$ for 10 s. Supernatants were transferred into a new tube containing 10 µl 0.5 M Tris-HCl and 1.5 M NaCl. The protein concentration was measured and the concentration was set to 50 µg per sample. DTT (Dithiothreitol, Gerbu, #1008.0025) was added to the samples to a final concentration of 5 mM and incubated at 25 °C for 1 h. CAA (Chloroacetamide) was added to each sample to a final concentration of 40 mM and incubated in the dark for 30 min. Last, trypsine was added to an enzyme:substrate ration of 1:75 and incubated at 25 °C overnight. Samples were purified using StageTip purification. The StageTips (Thermo Fisher, 13-110-061) were equilibrated using 20 µl 100% methanol, 20 µl 0.1% fomic acid in 80% acetonitrile, and 20 µl 0.1% formic acid in water. Between each step, StageTips were centrifuged at $800 \times g$ for 1 min. Afterward, the samples were centrifuged at full speed for 5 min. Next, the samples were loaded onto the StageTips. The StageTips were centrifuged at $800 \times g$ for 3 min. The tips were then washed first with 30 µl 0.1% formic acid in water and centrifuged at $800 \times g$ for 1 min and next twice with 30 µl 0.1 formic acid in 80% acetoniltrile. Tips were again centrifuged at $800 \times g$ for 1 min between washing steps. Samples were analyzed by the CECAD Proteomics Facility on an Orbitrap Exploris 480 equipped with FAIMSpro duo mass spectrometer that was coupled to an Vanquish neo in trap-and-elute setup (all Thermo Scientific). Samples were loaded onto a precolumn (Acclaim 5 µm PepMap

300 μ Cartridge) with a flow of 60 μl/min before reverse-flushed onto an in-house packed analytical column (30 cm length, 75 μm inner diameter, filled with 2.7 μm Poroshell EC120 C18, Agilent). Peptides were chromato-graphically separated with an initial flow rate of 400 nl/min and the following gradient: initial 2% B (0.1% formic acid in 80% acetonitrile), up to 6% in 3 min. Then, flow was reduced to 300 nl/min and B increased to 20% B in 26 min, up to 35% B within 15 min and up to 98% solvent B within 1 min while again increasing the flow to 400 nl/min, followed by column wash with 95% solvent B and re-equilibration to initial condition. The mass spectro-meter was operated in Top24 data-dependent acquisition with MS1 scans acquired from 350 m/z to 1400 m/z at 60k resolution and an AGC target of 300%. MS2 scans were acquired at 15 k resolution with a maximum injection time of 118 ms and a normalized AGC target of 50% in a 2 Th window and a fixed first mass of 110 m/z. All MS1 scans were stored as profile, all MS2 scans as centroid. All mass spectrometric raw data were processed with Maxquant (version 2.2[75],) using default parameters against a chimeric database con-sisting of the canonical Human UniProt database (UP5640, downloaded 04/01/2023) merged with 199 fetal bovine serum sequences[51]. Follow-up analysis was done in Perseus 1.6.15[76]. Protein groups were filtered for potential con-taminants and insecure identifications, and the Scube2-interacting protein candidates ranked according to the number of identified peptides. The mass spectrometry proteomics data have been deposited to the ProteomeXchange Consortium via the PRIDE partner repository with the dataset identifier PXD071371.

## Statistics and reproducibility
Data are expressed as means and error bars represent the standard devia-tions of the means. All datapoints are shown by dots and "n" in the figure legend represents the biological replicates of the group. All tests were per-formed and graphs were generated using GraphPad Prism (v10.6.1). Appropriate statistical tests (t-test, one-way or two-way ANOVA) and post hoc tests (Dunnett's multiple comparisons test or Sidaks multiple com-parisons test) are stated in the figure legends. A p-value of < 0.05 was considered statistically significant.

## Reporting summary
Further information on research design is available in the Nature Portfolio Reporting Summary linked to this article.

## Data availability
All data supporting the findings of this study can be found in this paper and its Supplementary Information. Supplementary_data_and_blots_COMMSBIO-25-0996B contains supporting experimental results, source data for all main figures, and also contains uncropped and unedited blot images. Source data are also provided with this paper as supplementary data. Supplementar-y_Data_1-COMMSBIO-25-0996B provides raw experimental data for Figs. 1–6, and Supplementary_Data_2-COMMSBIO-25-0996B shows Scube2-interacting proteome data. The proteome data are also available via ProteomeXchange with identifier PXD071371. All requests for materials and correspondence should be addressed to K.G.

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

## Acknowledgements
The excellent work of Petra Jakobs, Sabine Kupich, and Reiner Schulz is gratefully acknowledged. Scube2 constructs were kindly provided by Ming-Tzu Tsai and Ruey-Bing Yang (Academia Sinica, Teipei, Taiwan). Funding: This work was funded by the Deutsche Forschungsgemeinschaft (DFG, German Research Foundation)—Project-ID 386797833—SFB 1348: Dynamic Cellular Interfaces to K.G and S.V.W., and GR1748/9-1 to K.G. We thank the CECAD Proteomics Facility for the analysis of proteome data. This work was supported by the large instrument grant INST 216/1163-1 FUGG by the German Research Foundation (DFG Großgeräteantrag).

## Author contributions
K.G. and G.S. designed the studies, J.P., G.S., J.F., D.M., K.E., and J.W. performed the experiments, J.P., G.S., J.F., D.M., K.E., J.W., and K.G. carried out the analysis, C.G. provided materials and J.P., C.G., S.V.W., and K.G. drafted and revised the article.

## Funding

## Competing interests
The authors declare no competing interests.
