## [Transparent Peer Review file · Communications Biology]

Scube2 primes Dispatched and ADAM10-mediated Shh release by recruiting HDL acceptors to the plasma membrane

Corresponding Author: Professor Kay Grobe

Version 0:

Reviewer comments:

Reviewer #1

(Remarks to the Author)

Understanding hedgehog release and diffusion is critical in developmental biology and cellular signalling. The hedgehog (HH) signalling pathway plays a pivotal role in regulating tissue patterning, growth, and cell differentiation during embryonic development, as well as maintaining tissue homeostasis in adults. Hedgehog proteins, such as Sonic Hedgehog (Shh), must be released from producing cells and diffuse across tissues to establish concentration gradients that guide cellular responses. This process is further complicated by the post-translational modifications of hedgehog proteins, particularly the addition of cholesterol and palmitate lipids. These modifications are essential for their release, diffusion, and signal reception, but the mechanisms by which cells manage these lipid modifications remain an area of active research and debate. Multiple models have been proposed to explain the release and diffusion of hedgehog proteins, though consensus has yet to be reached. A key area of interest is understanding how these lipids are handled during the extraction of hedgehog proteins from producing cells, their movement through tissues, and their interaction with receptors. Previous studies, including by the authors, have highlighted the role of lipoprotein particles as carriers for lipidated hedgehog proteins. Additionally, the authors have previously demonstrated the involvement of Scube2 and ADAM10 in cleaving the lipidated termini from hedgehog molecules. Much of the work in this manuscript is confirmatory of these previous findings and is not particularly novel. The novelty comes from the identification of an interaction between Scube2 and apolipoprotein A1. The authors propose a model in which Scube2 helps to recruit lipoprotein particles to the cell surface bringing them into proximity of Adam10, Dispatched and Scube, which allows the transfer of cholesterol modified HH onto the LPPs. The authors reconfirm their previous published findings that such HH LPPs are biologically active. This work will be of significant interest to the field. The author's model, however, is contentious, as they propose that palmitate modification on Hedgehog (HH) is not essential for signaling, despite a substantial body of *in vivo* data in the literature demonstrating otherwise. Additionally, the crystal structure of Sonic Hedgehog (Shh) bound to the Patched receptor clearly illustrates the importance of palmitate for receptor engagement. Nevertheless, it is possible that multiple functional forms of HH, such as HH:LPP, may exist and play distinct roles under different conditions or exhibit unique properties in terms of diffusion range and signaling. Thus, understanding how HH is loaded onto Lipid-Protein Particles (LPPs) is crucial, and the work makes an important contribution to advancing our understanding in this area.

However, a significant limitation of this study, and indeed many studies on Hedgehog solubilization, lies in the reliance on HH secreted into the medium from tissue culture cells. This approach raises two concerns. First, it is unclear how well these findings recapitulate *in vivo* conditions. Second, it only allows for the analysis of HH that escapes the surface of the cells, potentially overlooking the most relevant pool of HH. The authors themselves demonstrate that Scube is highly adherent to heparan sulfate, as is HH, suggesting that the largest signaling-competent pool of HH may never leave the cell surface and is thus missed in the analysis. Additionally, the authors should expand their discussion to include other models in the literature, particularly those involving exosomes and cytonemes, which have also been implicated in Hedgehog release but are not addressed in this work.

Major comments.

- In solubilisation assays the authors only analyse the lower molecular weight cleaved form of HH. There is clearly also a higher molecular weight form (close in size to the form in/on the cells. What is this form and why is it ignored?

- Unpaired t-test, two-tailed is used for the statistics when quantifying HH release. This is not appropriate. When the control is set to 100% and variation is not analysed you cannot use an Unpaired two-tailed t-test. Because t-test compares the means of two independent groups and requires variability (or standard deviation) within each group.

- In all manipulations that perturb release of HH (such as A10, Dispatched mutations) where does the HH go? It is not going into the medium, but the amount recovered from the cells also does not appear to increase.

- Figure 3 (Drosophila work). This section is severely weakened by the indirect way HH release is analysed. The authors could and should look at what happens to HH protein itself when Kuz is absent. This could be done with clones' null for Kuz and an immunofluorescence-based analysis of what the impact is on HH release in the clone. The existing eye phenotypic analysis is also weakened by the failure to do a re-expression of wildtype HH as a positive control for the uncleavable HAHh expression.

- The Western blot in 5D is not of the same quality as others in the paper and probably is not sufficient quality for publication.

- The analysis on bioactivity of Shh/HDL really needs a comparison to dually lipidated Shh to see how active it is. Especially since previous work referenced (Palm, W. et al. Secretion and signaling activities of lipoprotein-associated hedgehog and non-sterol-modified hedgehog in flies and mammals. PLoS Biol 11, e1001505 (2013) shows that in vivo LPP associated Hh is not particularly active as it cannot activate high threshold target genes, which is contradictory to the authors claims of activity.

Reviewer #2

(Remarks to the Author)

Brief Summary:

In this manuscript, the authors identify a synergistic mechanism governing the release of Hedgehog (Hh) morphogens from the plasma membrane of the producing cell. The mechanisms of how Hh proteins are released to function as long range morphogens remains an ongoing and fundamental question to cell biology. Here, building off their recent previous work (PMID: 34308968, PMID: 39297609) the authors identify coordinated events between the Hh release proteins Dispatched1 (Disp), Scube2, A Disintegrin and Metalloproteinase 10 (A10), and Hh carriers like high-density lipoproteins (HDL). The authors results suggest Scube2 facilitates the transfer of Hh to lipoproteins (LPP) by binding to HDL via Scube2's EGF domains and simultaneously interacting with heparan sulfates on the cell surface. This work refines existing models of Hh release by developing a model to explaining how all these different known components may interact to facilitate release.

Overall impression of the work:

This work provides exciting new insights into mechanisms of Scube2 function and how it may interact with HDL and Shh. However, currently there is a lack of key controls and supporting experiments to justify many of the conclusions presented. Further experimentation is required before the conclusions presented can be justified.

Of note, the current format of this manuscript requires the groups previous results (Ehring et al 2024), to be well known to be able to accurately interpret several conclusions. Particularly with reference to Disp, as this is frequently not referenced correctly throughout the manuscript resulting in inaccurate claims. Many of the comments and concerns below may potentially be effectively resolved if this issue is considered.

Specific comments, with recommendations for addressing each comment:

Major points:

- 1) Figure S1: Better representative blots are required for accurate interpretation. Both A and B appear to have much higher protein levels of Scube2 in the HEK controls than the Disp^{-/-} or A10^{-/-} conditions, which would increase SHH release to the media.
- 2) Fig S1 B: lanes 1-4 it appears basal release of SHH is A10 independent. Can this be explained?
- 3) Figure 2: The addition of Disp^{-/-} cell analysis to the C25SShh, ShhN, and C25SShhN experiments may benefit the interpretation of the claims for Figure 2. Additionally, the corresponding text (Lines180-197) are difficult to interpret. In particular, "the release of truncated C25SShh was enhanced by A10 and Scube2 (Figure 2A, A', A''), but somewhat less A10 controlled than that of dual lipidated Shh (Figure 1C-C')." You do not analyze the loss of A10 alone. Further you are only adding Scube2, there is no addition of A10.
- 4) Figure 3: Reduced ommatidia may be caused by various signaling pathways, including Hh. It may be beneficial to further validate in developing tissue like the wing disc. Use of Hh-GAL4 or ap-Gal4 to drive UAS-KuzDN to look at Hh targets like Ptc and Ci, would confirm Hh pathway disruption. The authors may also wish to confirm reduction is not due to cell death.
- 5) For the authors consideration: How do the HDL levels used in these assays compare to levels in the developing and adult mouse and human body? E.g. A healthy human adult level of HDL = 60 mg/dl (600 ug/ml). Considering Scube2 mutants exhibit only mild phenotypes in mice, how and why is Scube2 necessary in mammalian systems? Perspectives and thoughts on this point may provide provocative ideas for the discussion based around the authors proposed model.
- 6) Figure 6F and S6E: The QCM-D chip experiment lacks the control of running Scube2 incubation alone in the absence of HDL, to see if there is a different Δ , confirming recruitment of HDL. It would also be beneficial to see this experiment

compared with using Scube2 Δ CUB and Scube2 Δ EGF to further validate the conclusions.

7) Figure S4A: Actin is present in both cell and media fractions. Shh release values from this blot cannot be interpreted.

8) Figure S4D: loading controls are required.

Minor Points:

9) Fig S1 D. loading controls (both actin and PonS) make the interpretation difficult.

10) Can the authors discuss how their model coincides or differs with other models showing Scube2 maintains interactions with SHH to interact with receptors on receiving cells, handing off SHH ligands to activate the pathway (PMID: 35231446)?

11) Line 234: Reference required for Hhbar3 mutation.

12) Line 286: Similar to line 193-195, "The specific requirement of Disp and Scube2 for A10-mediated Shh processing..." Disp requirement for A10 mediated SHH processing is not shown or referenced.

13) Line 193-195: "When one of the three components is missing, Shh release is severely reduced, suggesting that the three proteins cooperate in Shh release." Cooperation between A10 and Disp is not demonstrated here.

14) Line 369: "...we have previously rediscovered that Disp-dependent Shh release is strickly dependent on the presence of HDL in the medium." The phrase "previously rediscovered", is unclear in its meaning. This phrase is also used on line 522.

15) Figure S3B: Can a different blot be provided. The current blot makes interpretation of reduced SHH release difficult. Loading controls in lane 7 (A10 $^{-/-}$, Scube2 Δ CUB +) appear to have a reduced amounts of protein loaded in the media, as seen by the PonS, and corresponding large reduction in Scube2 levels impacting the amount of SHH.

16) Lines 317-321 provide an interesting point, but perhaps are more appropriate if moved to the discussion.

17) Figure 6A: Scube2-pull down assay. Although noted in line 794-798 it was not possible to detect HDL-loading controls by immunodetection in the input. Inclusion of the immunoblot of the elution would provide some insights into the input levels as an indirect loading control.

18) Figure S4C: expression data should include SD, or be represented as bar/scatter plots to show replicates.

19) Line 534-536: Rephrase this sentence. The authors are discussing Drosophila results which lack Scube2.

20) Line 715-720: Please explain in the methods why does the incubation times differ between 6 hours and 12 hours for the different conditions in the SHH release assays?

Version 1:

Reviewer comments:

Reviewer #1

(Remarks to the Author)

The authors have satisfactorily addressed all my concerns. I find certain aspects of the work particularly novel, such as the demonstration that Scube can recruit lipoprotein particles to the cell surface. The authors have also convinced me that their model is not entirely incompatible with previously published studies showing lipid engagement with Patched. However, I recommend that some of the points raised in the rebuttal on this topic be incorporated into the Discussion section. This would help provide readers with context regarding the relationship between the two models—for instance, whether lipids might access a binding pocket when added exogenously but not under physiological conditions, or how both mechanisms could operate concurrently, with their relative contributions modulating signaling intensity and range. Including such a discussion would help place the work in a broader and more balanced context for the reader.

Reviewer #2

(Remarks to the Author)

The authors have made substantial revisions, added new experiments, clarified the text, and improved the presentation of data. They have effectively addressed the initial concerns and enhanced the overall quality and rigor of the manuscript. In cases where specific points could not be addressed, they have provided clear and reasonable explanations. I believe the manuscript presents sufficient evidence to support the authors' conclusions, and I have no reservations about recommending it to move forward in the publication process.

COMMSBIO-25-0996-T Reply to reviewer's comments:

We would like to thank reviewers for their time, effort and their constructive criticism. In our resubmitted manuscript, we have addressed all points raised by the reviewers:

Reviewer #1 (Remarks to the Author):

Understanding hedgehog release and diffusion is critical in developmental biology and cellular signalling. The hedgehog (HH) signalling pathway plays a pivotal role in regulating tissue patterning, growth, and cell differentiation during embryonic development, as well as maintaining tissue homeostasis in adults. Hedgehog proteins, such as Sonic Hedgehog (Shh), must be released from producing cells and diffuse across tissues to establish concentration gradients that guide cellular responses. This process is further complicated by the post-translational modifications of hedgehog proteins, particularly the addition of cholesterol and palmitate lipids. These modifications are essential for their release, diffusion, and signal reception, but the mechanisms by which cells manage these lipid modifications remain an area of active research and debate. Multiple models have been proposed to explain the release and diffusion of hedgehog proteins, though consensus has yet to be reached. A key area of interest is understanding how these lipids are handled during the extraction of hedgehog proteins from producing cells, their movement through tissues, and their interaction with receptors. Previous studies, including by the authors, have highlighted the role of lipoprotein particles as carriers for lipidated hedgehog proteins. Additionally, the authors have previously demonstrated the involvement of Scube2 and ADAM10 in cleaving the lipidated termini from hedgehog molecules. Much of the work in this manuscript is confirmatory of these previous findings and is not particularly novel.

Author's comment: This reviewer correctly states that our group has previously provided RNAi-based evidence of ADAM10's involvement in Shh shedding and investigated LPP acceptor functions. However, the current study provides much stronger in vitro data using ADAM10 CRISPRd cells, and adds in vivo results on Kuzbanian-regulated Hh release. In the Hh field, the currently favored model of Hh solubilization and transport supports Dispatched- and Scube-mediated solubilization of dual-lipidated Shh, and rejects the idea that LPPs and ADAM10 play a role. Therefore, we believe that our data on Scube2-dependent ADAM10 activity in Shh processing and transfer to LPPs is important because it adds an alternative Hh release and transport model, rather than merely confirming our previous findings or the current model.

The novelty comes from the identification of an interaction between Scube2 and apolipoprotein A1. The authors propose a model in which Scube2 helps to recruit lipoprotein particles to the cell surface bringing them into proximity of Adam10, Dispatched and Scube, which allows the transfer of cholesterol modified HH onto the LPPs. The authors reconfirm their previous published findings that such HH LPPs are biologically active. This work will be of significant interest to the field. The author's model, however, is contentious, as they propose that palmitate modification on Hedgehog (HH) is not essential for signaling, despite a substantial body of in vivo data in the literature demonstrating otherwise. Additionally, the crystal structure of Sonic Hedgehog (Shh) bound to the Patched receptor clearly illustrates the importance of palmitate for receptor engagement.

Author's comment: This reviewer raises the most important point here. We are aware that the model we proposed in the past and refine in this study is perceived as being contentious because it suggests that the role of the highly conserved N-palmitate is to tether the Hh ligand to the cell surface until ADAM10-controlled cleavage releases the ligand, only then allowing for its C-cholesterol-dependent transfer to LPP acceptors. In contrast, the current Shh release and transport model states that N-palmitate contributes to Hh transport and signaling to the Ptc receptor and must therefore not be removed.

We would like to address the last remark of this reviewer first. First, we do not see a necessary clash between our data and the current model supported by the cryo-EM structure of Ptc with the Shh N-terminal palmitate inserted into the cholesterol transport tunnel¹, simply because more than one Hh transport and signaling mode likely exist. Moreover, this structure cannot definitively prove our model incorrect because it was artificially reconstituted using detergent-extracted, dual-lipidated Shh ligand. The central question arising from this approach is: Are Shh ligands solubilized under physiological conditions (i.e., by Dispatched and Scube2) also dual-lipidated? Thus far, no convincing direct demonstration of dual-lipidated Hh or Shh release has been published. Second, alternative cryo-EM Shh/Ptc structures can also explain Ptc activity regulation by Shh independently of the N-terminal palmitoylated peptide^{2,3}. Our past⁴ and current activity tests using N-processed or non-palmitoylated artificial C25AShh ligands suggest that these structures and signaling modes should not be*

ruled out. Thus, taken together, although the published structure by Qi et al. provides fascinating insight into a possible Shh signaling mode at the Ptc receptor, the strict interpretation of this data to exclude other signaling modes goes too far, in our view.

* This is with the exception of the study published by W. Palm and S. Eaton⁵. They used 3H cholesterol to demonstrate that one of the soluble, strongly bioactive Hh proteins they detected is deesteroylated – and wondered how. We note that their observation is consistent with our results obtained under serum-depleted conditions. Unfortunately, they did not publish the same assay using 3H palmitate. However, since their deesteroylated protein variant is monomeric in solution (Figure 3E in⁵), it must be assumed that it is also depalmitoylated — which would again be consistent with our results.

The second reviewer's comment relates to the importance of N-palmitate in Hh and Shh function in vivo. Proteins lacking these lipids show reduced activities (though not always!), which was interpreted as indicating that both lipids contribute to signaling at the level of the receptor, Ptc. The excitement for the above cryo-EM structure of X. Li's group stemmed from this Ptc/Shh structure finally supporting the widely held hypothesis with beautiful structural data. However, alternative explanations for the altered activities of engineered Hh/Shh ligands in vivo have never been considered. Indeed, a caveat of all publications is that they are based on ligands that lack N-palmitate, C-cholesterol, or both already during biosynthesis and at the cell surface. These monolipidated ligands are released in an spatiotemporally unregulated and Dispatched- and Scube2-independent manner (as shown in this and our previous study), which will also affect their biofunctions and better explains why signaling of engineered ligands can surprisingly be increased rather than decreased in some developmental contexts and tissues^{6,7}. Therefore, while our data supports the in vivo relevance of terminal Hh lipids, it provides another explanation for it.

Nevertheless, it is possible that multiple functional forms of HH, such as HH:LPP, may exist and play distinct roles under different conditions or exhibit unique properties in terms of diffusion range and signaling. Thus, understanding how HH is loaded onto Lipid-Protein Particles (LPPs) is crucial, and the work makes an important contribution to advancing our understanding in this area. However, a significant limitation of this study, and indeed many studies on Hedgehog solubilization, lies in the reliance on HH secreted into the medium from tissue culture cells. This approach raises two concerns. First, it is unclear how well these findings recapitulate in vivo conditions. Second, it only allows for the analysis of HH that escapes the surface of the cells, potentially overlooking the most relevant pool of HH. The authors themselves demonstrate that Scube is highly adherent to heparan sulfate, as is HH, suggesting that the largest signaling-competent pool of HH may never leave the cell surface and is thus missed in the analysis.

Author's comment: Once again, this reviewer makes an excellent point. Indeed, the currently accepted Shh transport model via Scube2 does not address the essential role of heparan sulfate (HS) in Hh biofunction, and the tight binding of Scube2 to HS on the surface of producing cells challenges the postulated mode of Scube2 chaperoning Hh away from the source (Figure 6 F-H, page 23 line 7 – 17). In our revised study, we demonstrate and discuss (page 23, lines 13 – 15, page 25, lines 11 – page 26, line 4) how the irreversible HS binding of Scube2 is not problematic for our model because its role is to recruit the HDL carrier for Shh to the release sites. The reviewer is correct that Shh also binds HS, which could impede solubilization in theory. However, we have previously demonstrated that Shh and Hh are highly mobile within the HS-rich matrix, a process made possible by direct switching of Hh ligands between HS chains. We have previously shown and discussed this^{8,9} (and cite this work, page 23 line 14). Therefore, this second concern is also not a problem for our model.

Regarding the first concern that our work relies on cell culture: we show in vivo data in this study (Figure 3, Supplementary Figure S2) that support our model. We address the second concern that the solubilized material may not leave the glycocalyx by side-by-side comparisons of cellular and solubilized material. This side-by-side comparison shows clearly that little to none truncated Shh (the bottom band) is retained at the cell surface (Please see Figures 1, 2, 5, and 6.).

Major comments

1) In solubilisation assays the authors only analyse the lower molecular weight cleaved form of HH. There is

clearly also a higher molecular weight form (close in size to the form in/on the cells. What is this form and why is it ignored?

Author's response: As demonstrated in this paper and a previous one ⁴, the "top" band in solution corresponds to a Shh fraction released independently of Dispatched, Scube2, and ADAM10. In contrast, the "bottom" truncated form only forms when all three release factors are present. Thus, we considered the latter to be the physiologically relevant form and excluded the former from further analysis, as it may be an overexpression artifact or represent the membrane-bound fraction after cell death.

Importantly, support for this interpretation can also be found in published data from Phil Beachy's group. Figure 6, shown in ¹⁰, depicts two Shh bands in Western blots of the medium. Only the bottom band is IPd when the function-blocking anti-Shh antibody 5E1 (that binds to the Ptc binding Shh epitope) is used. The top band remains in the medium and exhibits significantly reduced bioactivity. This is consistent with N-processing during release and the finding that only the processed protein is bioactive, as published by us before ¹¹.

2) Unpaired t-test, two-tailed is used for the statistics when quantifying HH release. This is not appropriate. When the control is set to 100% and variation is not analysed you cannot use an Unpaired two-tailed t-test. Because t-test compares the means of two independent groups and requires variability (or standard deviation) within each group.

Author's response: We fully agree and now use correlated t-tests on pairwise data that is not set to 100% in all figures.

3) In all manipulations that perturb release of HH (such as A10, Dispatched mutations) where does the HH go? It is not going into the medium, but the amount recovered from the cells also does not appear to increase.

Author's response: This is a commonly noted problem; therefore, we included an explanation in the Methods and Materials section of the revised manuscript (page 33, lines 9 ff.): "Note that the amounts of immunoblotted soluble and cellular Shh do not correlate inversely. This is because medium lanes represent all TCA-precipitated proteins, while cells were directly lysed in SDS buffer and only a small fraction (about 5%) were applied to the gel. As a consequence, a 100% increase in Shh solubilization will correlate to only 5% reduction in the amount of cell-surface-associated Shh, and vice versa."

4) Figure 3 (Drosophila work). This section is severely weakened by the indirect way HH release is analysed. The authors could and should look at what happens to HH protein itself when Kuz is absent. This could be done with clones' null for Kuz and an immunofluorescence-based analysis of what the impact is on HH release in the clone.

Author's response: We agree and conducted several analyses on Hh target gene expression in tissues that lack endogenous Kuz activity on the endogenous Hh ligand. These analyses support our end-point analysis. We continued using the eye disc because the use of several wing disc drivers resulted in early lethality, except for nubbin-Gal4-driven expression, which resulted in rare escapers with very small wings, similar to wing-specific Hh nulls. However, their extremely small wing discs prevented further analysis. In the eye, however, we supported our hypothesis of Kuz-regulated Hh solubilization by reduced Ci target gene expression at the disc (Figure 3C, Supplementary Figures 2B and 2C, page 11, lines 6ff).

5) The existing eye phenotypic analysis is also weakened by the failure to do a re-expression of wildtype HH as a positive control for the uncleavable HAHh expression.

We have also added a positive control for cleavage-deficient HAHh (see Figure 3B and 3B'). The beauty of this experimental approach to render the substrate sheddase-resistant is that it eliminates the pleiotropic effects of Kuz knockdown. Still, the effects of impaired HAHh cleavage on eye development closely resemble those of Disp, Kuz, or Hh knockdown, supporting our model.

6) The Western blot in 5D is not of the same quality as others in the paper and probably is not sufficient quality for publication.

Author's response: We fully agree and improved or replaced this and other lower-quality blot pictures.

7) The analysis on bioactivity of Shh/HDL really needs a comparison to dually lipidated Shh to see how active it is. Especially since previous work referenced (Palm, W. et al. Secretion and signaling activities of lipoprotein-associated hedgehog and non-sterol-modified hedgehog in flies and mammals. PLoS Biol 11, e1001505 (2013) shows that in vivo LPP associated Hh is not particularly active as it cannot activate high threshold target genes, which is contradictory to the authors claims of activity.

Author's response: We agree and repeated and extended the bioassays. We directly compared the bioactivities of the commercially available, dual-lipidated, artificial Shh variant used for the cryo-EM structure with Ptc and the truncated N-Shh variant linked to HDL, as requested. However, we noticed that the artificially detergent-solubilized Shh variant immediately associates with serum proteins when added to serum-containing media. This is not surprising, as both lipids find their thermodynamically favored environment, similar to what might occur during reconstitution with Ptc and its hydrophobic cholesterol-conducting tunnel prior to cryo-EM¹. We exploited this property by mixing equal amounts of the detergent-solubilized Shh ligand with the HDL also used for Shh release from cells (see Supplementary Figure 5D''). We then adjusted the protein levels (see Supplementary Figure 5D') and added equal amounts to the reporter cells. Interestingly, the activity of the dual-lipidated Shh ligand was reduced compared to the N-processed ligand (Figure 5D). We have previously explained this finding: The Ptc binding site is blocked by the uncleaved N-peptide. However, its proteolytic removal makes Shh soluble and also renders the Ptc (and 5E1 antibody) binding site more accessible¹¹.

We would also like to comment on the paper published by Palm and Eaton 12 years ago⁵. As is often the case, it is interesting to revisit old work and reread the manuscript with current data and concepts in mind. The original statement was that their new Shh/Hh variant lacking the C-sterol, called Hh-N/Shh-N*, is monomeric/dimeric and not LPP-associated, unlike most soluble proteins. They claimed that Hh-N* is more bioactive than LPP-associated Hh proteins in vivo. This is also stated in the abstract and is therefore best-remembered.*

Apart from the authors showing the opposite when using LPP:Shh proteins in vitro (Figure 7 in⁵), a few things are worth considering. First, the Hh proteins are overexpressed in the fat body, which is not a physiological system; they should have been produced in the wing disc. Second, interestingly, the Hh-N form not associated with LPPs is produced in vivo via coexpression of an LPP RNAi in the fat body in larvae, which depletes the carrier. Our in vitro data (in this study and in⁴) show that processing of the C-terminus occurs only when HDL levels are low, mirroring this condition of restricting LPP supply in larvae. More importantly, another problem of their LPP RNAi approach to generate Hh-N* and measure its activity in vivo is twofold: First, we now know that Ptc exports sterols to suppress Smoothened and that this export may depend on the availability of soluble sterol acceptors (i.e., LPPs). Figure 5 of their paper indeed shows this: LPP RNAi without Hh co-expression reduces LPP levels in the disc, inducing strong Ci accumulation in the entire anterior wing disc compartment and anterior overgrowth. The explanation now is that a lack of sterol acceptors impairs cholesterol export by Ptc, which the Hh ligand would also do using the same sterol transport-inhibition mechanism. LPP RNAi, in other words, alone already induces Smoothened and the Hh pathway strongly. Second, Supplementary Figure 5F shows that LPP RNAi induces the Hh-N* form but still releases LPP-associated Hh, producing a mix. On top, the fewer particles present carry more Hh per LPP (increasing signaling downstream of Ptc) than under non-LPP RNAi conditions, and also compete with fewer non-Hh-associated LPPs (suppressing signaling downstream of Ptc by acting as a sterol acceptor). This explains increased in vivo signaling independent of Hh-N* (which, by the way, is only moderately active in our assays⁴). Finally, their Shh release assays from HeLa cells were conducted in Scube2 absence, decreasing their processing.*

In summary: Examining the data retrospectively allows for a different interpretation. This reinterpretation leads to the exact opposite claim: LPP-associated Hh is bioactive, and concentrating it on fewer LPPs, together with impaired sterol export by Ptc by the limited availability of Hh-free LPP sterol acceptors, explains the increased in vivo signaling. This is not in conflict with, but rather supports, the results we present in our paper.

Reviewer #2 (Remarks to the Author):

Brief Summary: In this manuscript, the authors identify a synergistic mechanism governing the release of Hedgehog (Hh) morphogens from the plasma membrane of the producing cell. The mechanisms of how Hh proteins are released to function as long range morphogens remains an ongoing and fundamental question to cell biology. Here, building off their recent previous work (PMID: 34308968, PMID: 39297609) the authors identify coordinated events between the Hh release proteins Dispatched1 (Disp), Scube2, A Disintegrin and Metalloproteinase 10 (A10), and Hh carriers like high-density lipoproteins (HDL). The authors results suggest Scube2 facilitates the transfer of Hh to lipoproteins (LPP) by binding to HDL via Scube2's EGF domains and simultaneously interacting with heparan sulfates on the cell surface. This work refines existing models of Hh release by developing a model to explaining how all these different known components may interact to facilitate release.

Overall impression of the work: This work provides exciting new insights into mechanisms of Scube2 function and how it may interact with HDL and Shh. However, currently there is a lack of key controls and supporting experiments to justify many of the conclusions presented. Further experimentation is required before the conclusions presented can be justified.

Author's comment: We thank the reviewer for his positive overall evaluation. Regarding key controls and additional experiments, we have improved the revised manuscript according to the reviewer's suggestions, as outlined below.

Of note, the current format of this manuscript requires the groups previous results (Ehring et al 2024), to be well known to be able to accurately interpret several conclusions. Particularly with reference to Disp, as this is frequently not referenced correctly throughout the manuscript resulting in inaccurate claims. Many of the comments and concerns below may potentially be effectively resolved if this issue is considered.

Author's comment: We have also addressed this issue, as outlined below.

Specific comments, with recommendations for addressing each comment: Major points:

1) Figure S1: Better representative blots are required for accurate interpretation. Both A and B appear to have much higher protein levels of Scube2 in the HEK controls than the Disp^{-/-} or A10^{-/-} conditions, which would increase SHH release to the media.

Author's response: We have performed several additional release assays and improved protein expression assay controls for Supplementary Figure 1 A, B and D. Other improved blots from repeated additional experiments are shown in Figure 1C, Figure 5 c and D, and 6A.

2) Fig S1 B: lanes 1-4 it appears basal release of SHH is A10 independent. Can this be explained?

Author's response: We have not investigated this matter in detail, but we suspect that the combination of protein overexpression and the possibility that other sheddases compensate for the lack of ADAM10 may have caused this basal release. Nevertheless, we would like to point out that the presence of both ADAM10 and Scube2 increases Shh release by severalfold compared to its release in the absence of either ADAM10 or Scube2 (Figure 1C and Figure 5B).

3) Figure 2: The addition of Disp^{-/-} cell analysis to the C25SShh, ShhN, and C25SShhN experiments may benefit the interpretation of the claims for Figure 2. Additionally, the corresponding text (Lines180-197) are difficult to interpret. In particular, "the release of truncated C25SShh was enhanced by A10 and Scube2 (Figure 2A, A', A''), but somewhat less A10 controlled than that of dual lipidated Shh (Figure 1C-C')." You do not analyze the loss of A10 alone. Further you are only adding Scube2, there is no addition of A10.

Author's response: We presented the results in a previous publication. Therefore, to avoid redundancy and due to the space and word limits we must adhere to, we would prefer not to conduct these experiments again for the current submission. However, we now refer to these results in the main text (page 8, line 14ff). We have also

changed the wording of the above quoted sentence to: "We found that ^{C25S}Shh release was enhanced by A10 and Scube2 (Figure 2A, A', A"). The release of ShhN was not controlled by either A10 or Scube2 (Figure 2B' B"), suggesting that other sheddases can cleave the ShhN N-terminus under non-specific conditions (i.e. Shh lacking dual lipidation). Indeed, the level of ShhN solubilization was similar to that of ^{C25S}ShhN, an artificial non-lipidated control protein (Figure 2C-C", Supplementary Figure 1G)." (page 8, lines 8 - 13).

4) Figure 3: Reduced ommatidia may be caused by various signaling pathways, including Hh. It may be beneficial to further validate in developing tissue like the wing disc. Use of Hh-GAL4 or ap-Gal4 to drive UAS-KuzDN to look at Hh targets like Ptc and Ci, would confirm Hh pathway disruption.

Author's response: We conducted these suggested fly crossings and expressed kuzDN under the control of engrailed-Gal4, apterous-Gal4, Hh-Gal4 and nubbin-Gal4. However, with the exception of nubbin-Gal4, these crossings resulted in very early (embryonic or L1) lethality, even if conducted at 18°C (Supplementary Figure 1H). nubbin-Gal-4 driven KuzDN expression resulted in pupal lethality with rare escapers showing very small wings. However, the already extremely tiny discs in larvae prevented further analysis.

5) The authors may also wish to confirm reduction is not due to cell death.

Author's response: As shown in Supplementary Figure 2A, we now demonstrate that the small-eye phenotype is not due to increased apoptosis.

6) For the authors consideration: How do the HDL levels used in these assays compare to levels in the developing and adult mouse and human body? E.g. A healthy human adult level of HDL = 60 mg/dl (600 ug/ml). Considering Scube2 mutants exhibit only mild phenotypes in mice, how and why is Scube2 necessary in mammalian systems? Perspectives and thoughts on this point may provide provocative ideas for the discussion based around the authors proposed model.

Author's response: The concentration of LPPs in interstitial fluid is significantly lower than in plasma, with lymph/plasma concentration ratios of 0.03 for very low-density lipoproteins (VLDL) and 0.2 for HDL ¹². The relatively lower ratio for HDL is because interstitial fluid is a filtrate of blood serum that passes through capillary walls, therefore containing more small-mass LPPs from the serum ¹³. However, lymph can be more easily collected and HDL concentrations in lymph are therefore known, but are likely lower and vary depending on the tissue. In our revised manuscript, we have added an estimate of HDL in the lymph and how the levels used in our experiments relate to them: "To test this hypothesis, we solubilized Shh from HEK cells and A10^{-/-} cells at high (40 µg/ml, which represents about 30%-50% of the estimated HDL concentration in human lymph ¹³) extracellular HDL levels and in the presence or absence of Scube2." (page 20, lines 4-6).

7) Figure 6F and S6E: The QCM-D chip experiment lacks the control of running Scube2 incubation alone in the absence of HDL, to see if there is a different Δ , confirming recruitment of HDL. It would also be beneficial to see this experiment compared with using Scube2 Δ CUB and Scube2 Δ EGF to further validate the conclusions.

Author's response: We fully agree and have adapted the original QCM-D setup. We used the heparin-functionalized surface as before, but this time with purified Scube2 alone or pre-incubated with ApoA1 or HDL. We also conducted a QCM-D experiment in which we examined Scube2's interaction with the functionalized sensor surface in detail. Here, we extensively washed the bound protein with buffer to demonstrate that the interaction between Scube2 and heparin was very stable (Figure 6F). This was true even in the presence of soluble heparin in the wash buffer, demonstrating that Scube2 remains bound to surfaces functionalized with sulfated glycans. We support this property of Scube2 using flow cytometry (Supplementary Figure 7A). When Scube2 was pre-incubated with ApoA1, which alone showed negligible binding to heparin, we observed an increase in $-\Delta F$ and, most notably, an additional ΔD (Figure 6G). This supports the idea that the complex is also recruited to the functionalized sensor surface. While the results for HDL were less clear, we observed an increase in ΔD again supporting the recruitment of HDL to heparin (Figure 6H). However, these assays took the most time during our revision for two reasons: first, HDL is currently shipped with huge delays of several months; and second, the material we received was contaminated with factors that bound to the heparin surface even in

the absence of Scube2. Additionally, measuring the interaction between a large Scube2 complex and HDL under flow conditions was difficult due to the drag that weakened the interaction with the sensor. Nevertheless, the QCM-D assays presented in our revised manuscript, particularly those testing ApoA1, support the mechanism of Scube2-mediated HDL recruitment to the cell surface. We did not test Scube2 Δ CUB and Scube2 Δ EGF because IP experiments (Figure 6A and 6A') already demonstrated that only the EGF domains, but not the CUB domain, bind ApoA1 and HDL.

8) Figure S4A: Actin is present in both cell and media fractions. Shh release values from this blot cannot be interpreted.

Author's response: Due to the overload, we now show a 20kDa fragmentation product of 43kDa actin from the same blot picture as a normalization control. The control indicates that comparable cell lysis amounts were analyzed.

9) Figure S4D: loading controls are required.

Author's response: The Western blot, now Supplementary Figure 5C, shows the Shh loading controls for the bioactivity assay, demonstrating that similar amounts of Shh were present in the conditioned media and used in the differentiation assays. This requires no additional loading controls as shown for the Shh release blots.

Minor Points:

10) Fig S1 D. loading controls (both actin and PonS) make the interpretation difficult.

Author's response: We repeated the experiment and now show an improved result (Supplementary Figure 1D)

11) Can the authors discuss how their model coincides or differs with other models showing Scube2 maintains interactions with SHH to interact with receptors on receiving cells, handing off SHH ligands to activate the pathway (PMID: 35231446)?

Author's response: This is difficult and we refer this reviewer to our response to the remarks raised by Reviewer 1. As in the mentioned paper and its predecessor¹⁴, as well as the work of X. Li's group¹, the postulate is that Shh is transported in a dual-lipidated manner by Scube2 and handed off to Ptc via intermediates such as Gas1. Importantly, the N-palmitate is essential for both processes and for signaling to Ptc. These papers are based on the assumption that Shh and Hh are released in a dual-lipidated manner, but this was never shown directly and convincingly. Furthermore, neither group has attempted to express Shh in a defined, dual-lipidated manner for release characterization, as we do using our Shh/Hhat co-expression system. However, without quantitative N-palmitoylation, the N-terminal peptide will not be linked to the plasma membrane and will therefore not be a substrate for ADAMs, which cleave only membrane-linked peptides that are accessible in proximity to the membrane. In addition to our response to reviewer 1, we therefore believe the discrepancies between our results and those of the aforementioned groups can be traced back to different methodologies and quality controls. While not refuting the possibility of the hypothesized handover of dual-lipidated Shh, our past and present results clearly do not support this mechanism. We believe that discussing our different technical approaches, which would more or less directly criticize the experimental setups and lacking controls of others, should be avoided. We believe that most readers will understand the implications of our results without the need for further confrontative discussion.

12) Line 234: Reference required for Hhbar3 mutation.

Author's response: The original reference of the mutation from 1950 has now been added (Page 10, line 17).

13) Line 286: Similar to line 193-195, "The specific requirement of Disp and Scube2 for A10-mediated Shh processing..." Disp requirement for A10 mediated SHH processing is not shown or referenced.

Author's response: Following this reviewers suggestiuon, we changed this sentence to: "The specific requirement of Scube2 only for A10-mediated Shh processing provides an unexpected example of how the substrate specificity of an otherwise promiscuous protease can be effectively controlled,..." (page 13, line 10ff).

14) Line 193-195: "When one of the three components is missing, Shh release is severely reduced, suggesting that the three proteins cooperate in Shh release." Cooperation between A10 and Disp is not demonstrated here.

Author's response: This sentence was deleted from the revised manuscript.

15) Line 369: "...we have previously rediscovered that Disp-dependent Shh release is strickly dependent on the presence of HDL in the medium." The phrase "previously rediscovered", is unclear in its meaning. This phrase is also used on line 522.

Author's response: Following this reviewers suggestiuon, we rephrased this sentence to: "The second reason is that we have recently confirmed that Disp transfers Shh to HDL,..." (page 17, line 9ff).

16) Figure S3B: Can a different blot be provided. The current blot makes interpretation of reduced SHH release difficult. Loading controls in lane 7 (A10^{-/-}, Scube2 Δ CUB⁺) appear to have a reduced amounts of protein loaded in the media, as seen by the PonS, and corresponding large reduction in Scube2 levels impacting the amount of SHH.

Author's response: We have repeated the experiment and replaced Supplementary Figure 3B with the higher-quality result.

17) Lines 317-321 provide an interesting point, but perhaps are more appropriate if moved to the discussion.

Author's response: We agree that the functional aspect of Shifted EGF domains may also be appropriately placed in the Discussion section. However, we believe that the direct link of the EGF domains and the fact that the EGF domains of Scube2 are important (a new finding, as they were previously deemed nonessential for Hh release ¹⁰) make it a better fit in its current location. Therefore, we left this section unchanged.

18) Figure 6A: Scube2-pull down assay. Although noted in line 794-798 it was not possible to detect HDL-loading controls by immunodetection in the input. Inclusion of the immunoblot of the elution would provide some insights into the input levels as an indirect loading control.

Author's response: To address the reviewer's point, we reran the pull-down experiments and can now also demonstrate the formation of a precipitate from the input (Figure 6A, A'). Accordingly, we have changed the above quote from the Materials section to "HDL-input controls were obtained by protein precipitation and immunodetection of ApoA1 in the sample" (page 38 line 5-6).

19) Figure S4C: expression data should include SD, or be represented as bar/scatter plots to show replicates.

Author's response: The data is now presented according to this reviewer's suggestion (now shown in Figure S 5C).

20) Line 534-536: Rephrase this sentence. The authors are discussing Drosophila results which lack Scube2.

Author's response: This sentence was removed during the revision process when the manuscript was shortened and restructured.

21) Line 715-720: Please explain in the methods why does the incubation times differ between 6 hours and 12 hours for the different conditions in the SHH release assays?

Author's response: we have added the explanation to the materials section: "The longer incubation time was required due to the loss of cells during the washing steps and the corresponding reduction in released proteins per time and culture well" (page 34 line 3ff).

1. Qi, X., Schmiede, P., Coutavas, E., Wang, J. & Li, X. Structures of human Patched and its complex with native palmitoylated sonic hedgehog. *Nature* **560**, 128-132 (2018).
2. Gong, X. *et al.* Structural basis for the recognition of Sonic Hedgehog by human Patched1. *Science* **361**, eaas8935. (2018).
3. Qi, C., Di Minin, G., Vercellino, I., Wutz, A. & Korkhov, V.M. Structural basis of sterol recognition by human hedgehog receptor PTCH1. *Sci Adv* **5**, eaaw6490 (2019).
4. Ehring, K. *et al.* Two-way Dispatched function in Sonic hedgehog shedding and transfer to high-density lipoproteins. *Elife* **12** (2024).
5. Palm, W. *et al.* Secretion and signaling activities of lipoprotein-associated hedgehog and non-sterol-modified hedgehog in flies and mammals. *PLoS Biol* **11**, e1001505 (2013).
6. Lee, J.D. *et al.* An acylatable residue of Hedgehog is differentially required in *Drosophila* and mouse limb development. *Dev Biol* **233**, 122-136 (2001).
7. Li, Y., Zhang, H., Litingtung, Y. & Chiang, C. Cholesterol modification restricts the spread of Shh gradient in the limb bud. *Proc Natl Acad Sci U S A* **103**, 6548-6553 (2006).
8. Gude, F. *et al.* Hedgehog is relayed through dynamic heparan sulfate interactions to shape its gradient. *Nat Commun* **14**, 758 (2023).
9. Manikowski, D. *et al.* *Drosophila* hedgehog signaling range and robustness depend on direct and sustained heparan sulfate interactions. *Front. Mol. Biosci.* **10**:1130064 (2023).
10. Creanga, A. *et al.* Scube/You activity mediates release of dually lipid-modified Hedgehog signal in soluble form. *Genes Dev* **26**, 1312-1325 (2012).
11. Ohlig, S. *et al.* Sonic hedgehog shedding results in functional activation of the solubilized protein. *Dev Cell* **20**, 764-774 (2011).
12. Sloop, C.H., Dory, L. & Roheim, P.S. Interstitial fluid lipoproteins. *J Lipid Res* **28**, 225-237 (1987).
13. Lundberg, J., Rudling, M. & Angelin, B. Interstitial fluid lipoproteins. *Curr Opin Lipidol* **24**, 327-331 (2013).
14. Wierbowski, B.M. *et al.* Hedgehog Pathway Activation Requires Coreceptor-Catalyzed, Lipid-Dependent Relay of the Sonic Hedgehog Ligand. *Dev Cell* **55**, 450-467 e458 (2020).